# Statistical Modelling of Sediment Supply in Torrent Catchments of the Northern French Alps

Maxime Morel[1,2], Guillaume Piton[1], Damien Kuss[3,4], Guillaume Evin[1], and Caroline Le Bouteiller[1]

[1]Univ. Grenoble Alpes, INRAE, CNRS, IRD, Grenoble INP, IGE, 38000 Grenoble, France

[2]Office National des Forêts - service de Restauration des Terrains en Montagne des Hautes-Alpes, 5 rue des Silos, 05000 Gap, France

[3]Office National des Forêts - Direction Forêt & Risques Naturels, 9 quai Créqui, 38000 Grenoble, France

[4]Syndicat Mixte des Bassins Hydrauliques de l'Isère, 9 rue Jean Bocq, 38022 Grenoble, France

**Correspondence:** Guillaume PITON (guillaume.piton@inrae.fr)

**Abstract.**

The ability to understand and predict coarse sediment transport in torrent catchments is a key element for the protection and prevention against the associated hazards. In this study, we collected data describing sediment supply at 99 torrential catchments in the Northern French Alps. The sample covers a wide range of geomorphic activity: from torrents experiencing debris flows every few years to fully forested catchments exporting small bedload volumes every decade. These catchments have long records of past events and sediment supply to debris basins. The mean annual, the 10-year return period and the reference volume (i.e. the 100-year return level or the largest observed volume) of sediment supply were derived for studied torrents. We examined the relationships between specific sediment supply volumes and many explanatory variables using linear regression and random forest approaches. Results showed that the ratio of sediment contributing area (bare soil) to catchment area is the most important predictor of the sediment production specific volumes ($m^3/km^2$). Others variables such as the Metlon index or the indices of sediment connectivity have also an influence. Several predictive models were developed in order to estimate the sediment supply in torrents that are not equipped with debris basins.

## 1 Introduction

In mountain areas, the knowledge of the mean annual and event-driven sediment supply potential is important for the assessment of torrential hazards and the management of torrent catchments. Four main groups of approaches are typically employed to predict the volumes produced by debris flows and/or floods: (1) empirical approaches relating volume to catchment descrip-

tive parameters (e.g. Takei, 1984; Marchi and D'Agostino, 2004); (2) hydrological approaches considering the link between volumes and water flows (Rickenmann and Koschni, 2010); (3) geomorphological approaches estimating volumes from on-site recognition of sediment sources located along the channel network (Hungr et al., 1984); and (4) historical approaches assessing

volumes from data observed during previous events (e.g. test pits, topographic surveys of the deposited volumes, dredging of debris basin - D'Agostino, 2013). If there is sufficient data, a frequency analysis can also be considered (Jakob and Friele, 2010).

Empirical methods are relatively simple approaches to estimate the material supply of a torrent and are commonly used in engineering projects (Jakob, 2021). Several relationships have been established in the literature (Table 1) involving only one

or multiple parameters with, for example, a surface area parameter (catchment area, sediment contributing area also called effective catchment area or contributive catchment area - Harvey, 2002; Fryirs, 2013), a slope parameter (stream slope, fan slope), a length parameter (length of the erodible channel) or a parameter relating to the geological/geomorphological context reflecting the potential for erodible materials. It is also worth noting that these relationships have been produced from regression (i.e. mean trends) or envelope curve models (i.e. maximum potential).

When sufficiently data are available, the 10- and 100-year volumes can be calculated using a frequency analysis similar to that used for example in hydrology. D'Agostino and Marchi (2001) for instance proposed an envelope curve parametrized with the return period based on data coming from only three torrents. Time series of event magnitude are seldom available in torrent catchments. As can be seen Table 1, the existing relationships are mostly related to event-based data sets (where each event observation corresponds to a different catchment), and for which the prediction cannot be linked to any notion of return period.

Furthermore, these relationships have generally been calibrated on a limited number of torrents and/or with short observation periods. Interesting trends can nonetheless be capture on small samples: the envelope curve $V = 70,000 \cdot A$ that was eye-fitted by D'Agostino and Marchi (2001) on 84 events is for instance very close from the quantile equation $V_{99\%} = 77,000 \cdot A^{1.01}$ proposed by Marchi et al. (2019) for the same region on a ten time larger dataset. Meanwhile, some of these approaches are specifically focused on debris flows and/or have been calibrated on "specific" torrents, i.e. particular, very active torrents

producing large amounts of sediment. Since most empirical equations were derived from samples of very active torrents, these equations may lead to overly conservative estimations when applied to less active catchments. Indeed, these less active, dormant torrents produce very erratically large amounts of sediment and their low background sediment production is usually unknown. Luckily, some of these rarely active catchments in the French Alps were also equipped with debris basins.

Why sediment export varies so much between mountain catchments remains a very active research topic. In addition to

45 catchment size, slope, land cover and geology, the sediment connectivity is increasingly highlighted as a key driving factor (Heckmann et al., 2018; Altmann et al., 2021; Arabkhedri et al., 2021). Recently, the concept of (sediment) connectivity has been introduced to describe the efficiency of sediment transfer from its sources to the river system and the links between sources and sinks of sediment such as lakes in the upstream areas of catchments (Fryirs, 2013). The connectivity and disconnectivity of sediment sources to river systems within catchments are essential for sediment fluxes and thus for sediment export at

50 catchment outlets. Several indices have been developed to quantify this phenomenon in the catchments (Heckmann et al.,

2018). In particular, the index of connectivity (IC) (Borselli et al., 2008; Cavalli et al., 2013) has been widely used in torrential catchment studies (e.g. Micheletti and Lane, 2016; Blanpied et al., 2018; Schopper et al., 2019).

Table 1: Summary of the main empirical methods for predicting sediment production volumes of Envelope Curve type (EC) or regression equation (RE). See definition of the parameters in notation list in the appendix.

| Reference | Equation | Type | Description |
|---|---|---|---|
| Zeller (1976) | $V_{100} = 755.1 \cdot \chi_1 \cdot A^{0,782}$ | ? | Equation based on a sample of 16 events that occurred in Uri region (Switzerland). Area of catchments between 1 ha and 10 km$^2$. Values of coefficient $\chi_1$ for a "100-year" event between 6 and 20 |
| Takei (1984) | $V = 13,600 \cdot \chi_2 \cdot A^{0.61}$ | RE | Equations based on 551 events in Japan between 1972 and 1977. $\chi_2 = 1.0$ correspond to a median regression, other values can be considered to get higher estimates, e.g. $\chi_2 = 5.22$ or 8.38 correspond to an envelop curve of 90% and 95% of data, respectively |
| Kronfellner-Kraus (1984) | $V = 100 \cdot \chi_3 \cdot A \cdot S$ | ? | The author specifies that this formula gives the order of magnitude of the volume during extreme events. Relationship established using a sample of 1420 cases in the Austrian Alps. $\chi_3$ varies between 250 and 1750 for low to very active basins |
| Rickenmann (1997) | $V = \chi_5 \cdot A^{0.78}$ $(a)$<br>$V = (640 \cdot S_C - 23) \cdot L$ $(b)$<br>$V = (110 - 250 \cdot S_C) \cdot L$ $(c)$ | EC | Equations given in the case of a very high solid transport. $(a)$ : $\chi_5$ range from 17,000 to 27,000. It is considered here that the relation with a coefficient of 27 000 is an envelope curve. $(b)$ : if 0.07 m/m $\leq S_C \leq$ 0.15 m/m. $(c)$ : if 0.15 m/m $\leq S_C \leq$ 0.40 m/m |
| D'Agostino and Marchi (2001) | $V = 70,000 \cdot A$ | EC | Relationship established on a sample of 84 cases in the upper part of the Adige basin (province of Bolzano, Italy). |
| | $V = 25,415 \cdot A \cdot S^{1.28} \cdot GI$ | RE | The sample includes very diverse data with volumes ranging from 700 m$^3$ to 950,000 m$^3$ |
| | $V_T = (14,500 \cdot (-\log(-\log(1-1/T)) - 3,500) \cdot A$ | EC | Envelope curve built from an analysis on 3 Italian catchments: Moscardo torrent (4.1 km$^2$, 9 years of data), Rio Bianco torrent (0.88 km$^2$, 20 years of data) and Rio Inferno torrent (0.69 km$^2$, 46 years of data) |

| Reference | Equation | Type | Description |
|---|---|---|---|
| Franzi and Bianco (2001) | $V = 8,959 \cdot A^{0.7652}$ | RE | Equation based on a sample of 201 data from the Swiss, Italian and French Alps |
| Marchi and D'Agostino (2004) | $V = 65,000 \cdot A^{1.35} \cdot S^{1.7}$ <br> $V = 18,000 \cdot A^{1.16} \cdot S^{1.3} \cdot GI$ | RE | Same approach as that undertaken in the work of D'Agostino et al. (1996) and Marchi and D'Agostino (2004) with an expanded sample: 125 events in Italy in the eastern part of the Italian Alps, a doubling of the sample compared to 1996. But the statistical analysis that led to the relationships was performed on 86 cases where $V/A > 2,500$ m$^3$/km$^2$ and $V/L > 2,1$ m$^3$/ml |
| Peteuil et al. | $V = 25,000 \cdot A$ <br> $V = 120,000 \cdot (A \cdot R_{zp})^{0.6}$ | EC | Relationships resulting from a sample of 72 torrential catchments mainly located in the french northern Alps |
| Marchi et al. (2019) | $V_{50\%} = (2,620 \pm 60) \cdot A^{0.67 \pm 0.02}$ <br> $V_{99\%} = (77,000 \pm 7,000) \cdot A^{1.01 \pm 0.06}$ | RE | Quantiles 50% and 99% of 808 debris-flow volumes that occurred in 537 basins of the Eastern Italian Alps, basins with 97% of catchment areas <10 km$^2$ and all <33 km$^2$ (Marchi and Crema, 2018) |

This study aims to present a new prediction approach overcoming the several limitations pointed above. It is based on multivariate statistical models calibrated from an original data set covering 99 torrent catchments in the Northern French Alps. These catchments have a wide spectrum of sediment contributing area and several order of magnitude of sediment specific yield, and their sediment supply records have been documented for years to decades. From these records we were able to estimate statistically the mean annual, 10-year return period, and reference volume. Then we examined the relation between theses volumes and several explanatory variables such as geomorphological, hydro-climatic, geological or sediment connectivity indicators. Statistical approaches (random forest and power law regressions) were used for a simple use based on the most significant indicators. The paper first presents the study area, the selection of the sites and how can be extracted the explanatory variables. It secondly explores the correlation between catchment scale parameter and exported sediment volumes. The accuracy, application domain and limitations of the method developed are finally discussed.

## 2 Materials and methods

### 2.1 Study area

The study area is located in the northern french Alps. The studied catchments are located on a wide range of mountain setting, from hills culminating below 800 m.a.s.l. at the north-west of Grenoble to torrents draining the glaciers of the Chamonix valley

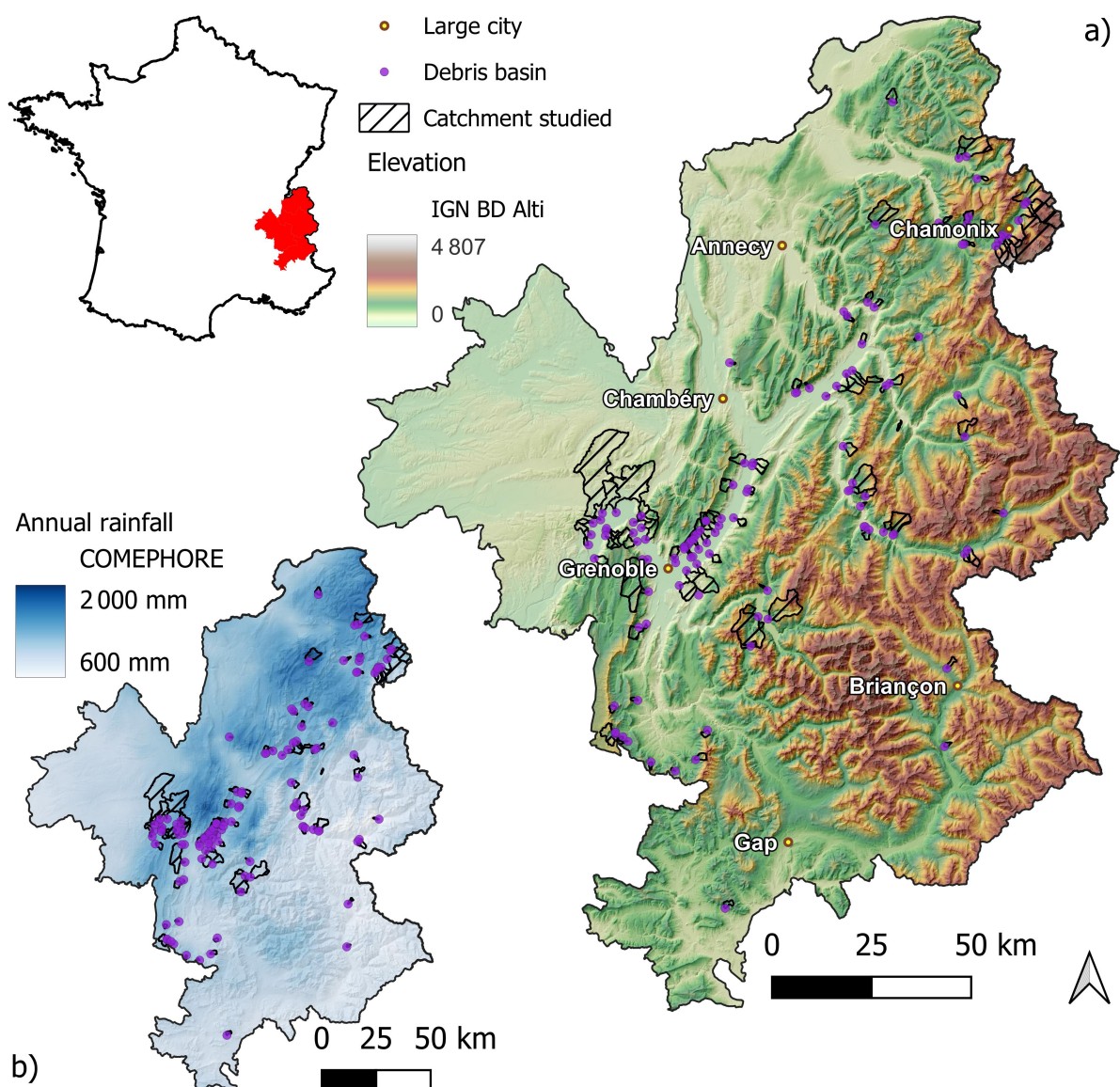

**Figure 1.** Spatial distribution of the studied sites: a) background image of elevation according to the IGN BD ALTI dabase and b) background image of mean annual rainfall according to the COMEPHORE data base (a link to access maps of each catchment is provided at the end of the paper).

with summits above 4,000 m.a.s.l. (Figure 1a). The geology of the studied catchments cover both sedimentary, metamorphic and igneous rocks. The climate in the area is considered temperate without dry summer in the valleys, usually cold without dry summer above 1000 m.a.s.l. and even polar above 2000 m.a.s.l. (sensu. Beck et al., 2018). The annual mean precipitation ranges within 600 and 2000 mm with a clear influence of the relief, as well as a decreasing trend toward the east (Figure 1b) associated to the penetration into the massif of the humidity coming from the Atlantic sea.

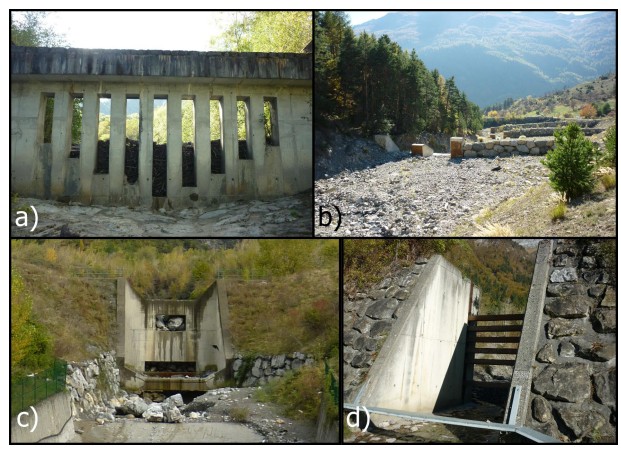

**Figure 2.** Examples of debris basins used in the analysis: a) Lavanchon torrent (Saint-Paul-de-Varces community), b) Verdarel torrent (Saint-Chaffrey community), c) Claret torrent (Saint-Julien-Mont-Denis community) and d) Nant Croex (Ugine community).

## 2.2 Sediment yield data

Data on sediment volumes were collected from the monitoring of debris basin dredging at the outlet of 120 investigated catchments. Those structures are managed by the french torrent control service ONF-RTM (*Office National des Forêts - Restauration des Terrains en Montagne*, n = 44 debris basins) or by local stakeholders such as river managers or municipalities (n = 76 debris basins). Structures managed by ONF-RTM concern generally the most active catchments which were acquired by the French government during the late $19^{th}$ and early $20^{th}$ centuries to perform reforestation and torrent control masterplans (Piton et al., 2017). In these active catchments, debris basins were later built to protect various assets (e.g. roads, urbanized areas). Debris basins managed by local stakeholders concern mostly small and less productive catchments that nonetheless experience erratic intense bedload transport during strong floods. The types and shapes of the debris basin are heterogeneous (e.g. Figure 2). The inventory of Carladous et al. (2022) showed that most of these structures, despite their variety, trap most of the coarse sediment supplied: their dredging is thus a relevant proxy of the catchment sediment yields. These structures are generally located near the apex of an alluvial fan, but some of them are located in other parts of the catchment to protect specific assets. Depending on the activity of the catchment, dredging was carried out several times a year or as soon as the structure was partially filled. The dredging operations were mainly carried out using mechanical excavators and trucks. Estimates of volumes were generally made by counting the trucks evacuating the sediment and at lesser times by comparing topographic surveys. It is noteworthy that these measurements have uncertainties that are difficult to quantify, but that we assume to be $\pm\ 25\%$ from expert knowledge. Due to the regular dredging of the debris basins, data could be collected continuously for each studied catchment, dredging operation covered periods ranging from 5 to 40 years (mean = 25 years). A total of 797 dredging operations were recorded (with a mean number per catchment of 7, and a range of 1 to 28).

In addition, another source of sediment supply data in the 120 basins comes from the national database on torrential events (BD-RTM; https://bdrtm.onf.fr). Briefly, this database provides information on past events that triggered damages in the catch-

ments (to protection structures, roads or buildings), giving details of any causes, eventually information on the process type
(i.e. debris flow or flood) and sometime also the volumes of sediment transported. This database is complementary to the debris basin dredging data as it provides data on sediment supply that occurred before the structures were built. Moreover, the event volumes are sometimes more reliable in cases where the sediment volume is higher than the capacity of the structure, since they include an estimation of all the deposits. The BD-RTM provided details on 348 events in the studied catchments. The number of events per catchment varies from 0 to 19. Similarly to dredging volumes, estimates of these volumes are subject to
uncertainties that are difficult to quantify.

A check of the input data was applied especially when an event occurred in the same year as a dredging operation. In the case of inconsistent estimated volumes from both events and dredging, we made a correction by retaining the largest volume. This ensures a safe estimate of sediment production in the catchment.

## 2.3    Estimation of sediment yield characteristics

Average annual sediment yields were estimated by analyzing the sediment supply data within a temporal window corresponding to the period of monitoring of the debris basin dredging (i.e., from the year of the structure construction to the most recent year for which the managers provided dredging operation information). We estimated the mean annual production $V_m$ by taking into account the volumes of events and dredging volumes (data from the managers of the structure and the BD-RTM). The data check identified several suspicious structures with clear change in dredging frequency suggesting a change in the function
of the structure (e.g., when cleaning is abandoned, the structure tends to maintain a sediment regulation zone where flow spread freely and pseudo-cycles of erosion and deposition occurs freely). These structures can lead to an incorrect estimation of average volumes and were excluded from the data set. In the end, 99 structures were retained for the analysis.

For the torrents where long enough records were available, individual frequency analyses for each torrent were performed to estimate the quantile representing the sediment supply volume for a 10-year return period $V_{10}$, as well as the reference
volume $V_{ref}$ (Fig. 3). The latter refers either to the volume of the largest known and documented event (about 20% of the sample), or to a theoretical 100-year return period event, if higher than the largest known event as was the case for about 80% of the catchments. Generalized Pareto distribution (GPD) or exponential type adjustments were performed depending on the number of observations (Coles, 2001). The data set was separated into three sub-samples based on the number of non-null and unique observations of sediment volume $n$ : (1) if $n \leq 5$ (50 % of the cases), we computed the mean annual production
but avoid extrapolation. These watersheds were therefore not used to estimate $V_{10}$ and $V_{100}$; (2) if $5 \leq n < 10$ (25 % of the cases), an exponential distribution was adjusted ; (3) if $n \geq 10$ observations (25 % of the cases), a GPD distribution was adjusted. Finally, we estimated $V_{10}$ and $V_{ref}$ for 69 catchments. The plots of the reconstructed sediment yield time series and the statistical adjustments are presented for the individual catchments in the Figure S1.

The volumes were also normalized by the watershed area $A$ to work with specific sediment yields ($V_m/A$, $V_{10}/A$ and
$V_{ref}/A$) expressed in m$^3$/km$^2$ that can more easily be cross-compared between catchments.

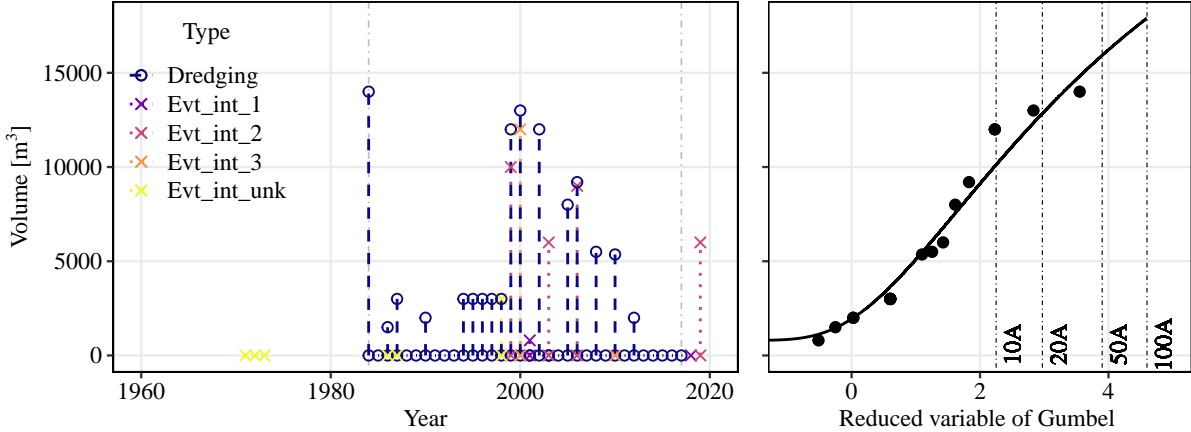

**Figure 3.** Example of a reconstructed sediment yield time series: case of the Arches torrent (Isère, France). The left panel represents the amount of solid input volume and the years of occurrence of these inputs. The types of dots and lines differentiate the type of input data (i.e., dredging data or historical event). The color of the dots and lines informs the intensity associated with the historical event ("Evt_int_1", "Evt_int_2", "Evt_int_3", "Evt_int_unk" refer to minor, moderate, high and unknown intensity, respectively). The two vertical dashed lines in grey delineate the monitoring period. The right panel shows the adjustment performed from the observations to estimate the volumes. The monitoring period covers 35 years and presents 19 years with non-null observations. A GPD distribution was adjusted.

## 2.4 Explanatory variables of sediment yield

Inspired by our expert knowledge, by previous works cited in Table 1 and by the literature on sediment connectivity, we did our best to estimate simple proxies of potential drivers of the sediment production: rainfall, relief, geology, land use, connectivity indexes and process type. More sophisticated indicators and models certainly exist but using them on a sample of about one hundred catchments was out of the scope of this work which was to develop a method simple to use.

### 2.4.1 Precipitation

An analysis of the rainfall chronicles was carried out for each catchment in order to obtain rainfall quantile values. For this purpose, the rainfall data from the COMEPHORE reanalysis were used (©Météo France, see https://radarsmf.aeris-data.fr/). These data provide precipitation values at an hourly time interval, on a 1 km$^2$ resolution grid and over the period 1997-2017. The COMEPHORE product exploits ground measurements from rain gauges and radars. It is considered to represent adequately the spatial extent and intensity of intense and local precipitation events (see Appendix A in Caillaud et al., 2021, for an extensive description of its strengths and limitations). To have a single hourly value in each catchment, weighted averages of the set of COMEPHORE grid cells included in the catchment extent were computed. The three annual maximum rainfall events of 1 h, 6 h and 24 h duration were extracted from the time series and their empirical probability of occurrence was estimated using the Weibull formula (Coles, 2001). The 10-year return period values of each rainfall duration was then computed ($P1h_{10}$, $P6h_{10}$, and $P24h_{10}$, respectively).

### 2.4.2 Morphometric parameters

The Melton index $M$ is an index of the ruggedness of the catchment (Melton, 1965) and is calculated as the ratio of the catchment relief (difference between the catchment maximal elevation and the elevation of the debris basin) by the square root of the catchment area measured at the debris basin. It is a normalized index of the gravitational energy of the catchment.

The mean stream slope $S_{CE}$ and the fan slope $S_C$ were calculated. The first refers to the mean slope of the reach located upstream of the debris basin and controlling the sediment transport, i.e. the reach with the mildest slope. The latter refers to the mean slope of the reach at the fan apex measured along a length equal to $10 - 20$ channel widths (Bertrand et al., 2013).

All variables were calculated using the 25 m resolution national DTM covering the entire study area (BD Alti ®, see details in https://geoservices.ign.fr/bdalti and Discussion about the related uncertainties).

### 2.4.3 Sediment transport processes

Several studies revealed that it is possible to discriminate catchments depending on the dominant sediment transport process using geomorphic characteristics (although these methods cannot be very accurate, they provide a first approximation, see Church and Jakob, 2020). Wilford et al. (2004) developed a method using the stream length combined with the Melton index to differentiate between debris-flow-prone, debris flood-prone and flood-prone catchments. Bertrand et al. (2013) developed a model to discriminate debris-flow- and flood-prone torrents using Melton index and fan slope. Examination of the watershed classes according to the two methodologies shows that the method of Bertrand et al. (2013) tends to merge debris flows and debris floods in the same pool (Figure 4). This may be due to the estimation of the fan slope which can be uncertain because of the coarse resolution of the DTM and because some catchments do not have a clearly defined fan.

For these reasons, we adopted the method of Wilford et al. (2004) for the study. Only this automatic classification was used without exhaustive cross-checking with field evidences due to the lack of availability of relevant and rigorous documentation on this question. In addition, many catchments experience mixed regimes where frequent and small events are rather related to bedload transport while infrequent, larger events might be debris flows (e.g. Theule et al., 2012; Marchi and Cavalli, 2007; Hübl, 2018): assigning a category is thus challenging. We decided to use the simple classification proposed by Wilford et al. (2004) – which is straightforward to use even on a undocumented catchment – simply to test if these classes emerged as sub-samples having clearly different sediment production capacities. It must be acknowledge that this is only a simplistic indicator and not a field-based evidence of a flow process type.

### 2.4.4 Sediment contributing area: connected eroding areas

We delineated the sediment contributing area using the French GIS database of the forest inventory as a mask (BD Forêt® V2, mapping vegetation units of surface $> 5000$ m$^2$, see https://geoservices.ign.fr/bdforet). This database provides an accurate digitization of the different natural vegetation covers (with information of the vegetation type, e.g. moorland, herbaceous formations, deciduous forest, coniferous forests). Land without vegetation cover can correspond to agricultural areas, to artificial areas (e.g. urban, road) or to bare soil – i.e. unvegetated soil, sediment or rock – often located in the headwaters. Here, areas

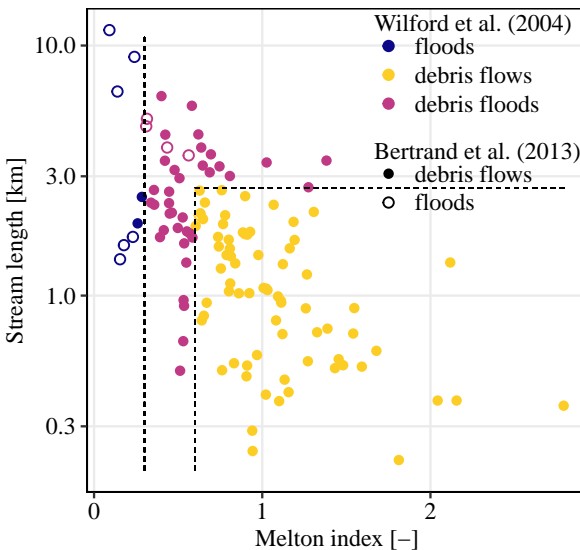

**Figure 4.** Catchment classifications based on the dominant sediment transport process. The color of the points indicates the classification according to Wilford et al. (2004). The dashed lines indicate the discriminating limits of the different categories according to this method. The type of dot indicates the classification according to Bertrand et al. (2013).

with bare soil were considered as potentially sediment contributing area. The definition is thus not exactly the same than that
used by Haas et al. (2011) and Altmann et al. (2021) who used automated threshold conditions on the land cover, the hillslope
gradient, the distance to the channel and the channel slope and but goes essentially in the same spirit: identifying in mountain
catchments connected, active sediment sources on aerial pictures – to identify the bare soil – and topographical maps – to check
the connectivity. To some extent, vegetated areas considered as moorland may sometimes be potentially sediment-producing
diffuse gullying areas. These areas were included or excluded from the potential sediment contributing area after a quick vi-
sual assessment of each area. It is worth mentioning that bare bedrock is also included in the sediment contributing area in
our approach. Although bedrock also produce sediment, bare soil has usually a much higher erosion rate. Any surface area
of connected bare soil or rock is however considered equally in our approach, their lithological differences is assessed in the
Geological Index presented in the next sub-section.

We then characterized if these areas were connected to the channel network (Figure 5). For this purpose, following Peteuil
and Liébault (2011), an area of bare soil was considered connected and thus part of the sediment contributing area if (i)
continuous bare soil was visible between the hillslope and the channel on the orthophotos, or (ii) a permanent or ephemeral
watercourse draining this area was present in the BD TOPO® database (base of detailed hiking maps usable down to a scale of
1:2000, see https://geoservices.ign.fr/bdtopo). Bare soil areas located upstream lakes or glaciers were considered disconnected.

The data processing allowed us to calculate the connected sediment contributing area for each catchment ($A_{ZP}$). We are
thereafter interested in the proportion of sediment contributing area to catchment area $R_{ZP} = A_{ZP}/A$.

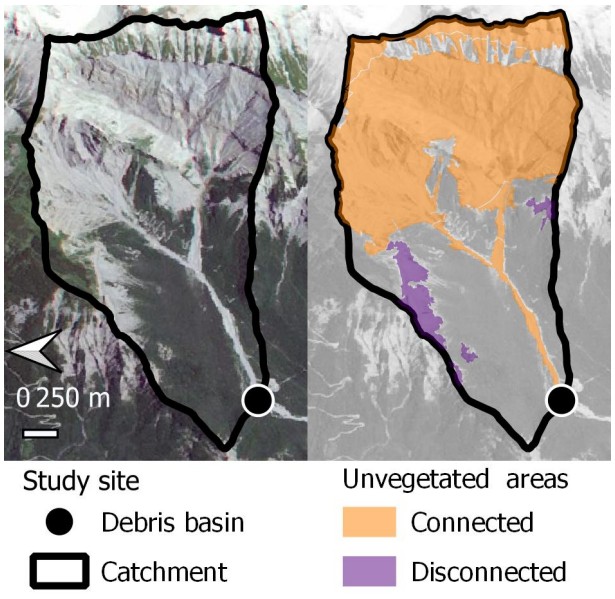

**Figure 5.** Delineation of sediment contributing area, example of the Ebron torrent catchment (Isère, France). Most of the bare soil is connected (orange polygon) but two disconnected talus slopes without vegetation can be seen on each side of the main channel (purple polygons) and are not considered in the computation of the sediment contributing area (background image source: RapidEye 2010).

We identified 37 catchments for which no bare soil area was delineated with our protocol. Having catchments with null value as sediment contributing area would be an issue for the statistical method used. After analysis of aerial images, these catchments do not have clearly visible sediment contributing areas, except for two watersheds for which the BD Forêt® does not detect small bare soil zones. For these catchments, we assumed that the very small sources of sediment are essentially remobilization of sediment from the stream bed (Anderson, 1949). We approximated the channel widths from hydraulic geometry relationships that relate the bankfull width to the catchment area. Hydraulic geometry models have been recently developed in France on a national and regional scale (Gob et al., 2014). In our case, we estimated the bankfull width $W_{bf}$ for all study sites using the regional hydraulic geometry model built from observations in the inner French Alps: $W_{bf} = 5.06 \cdot A^{0.27}$. Channel bed area was estimated by the product $W_{pb} \cdot L_{CE}$, where $L_{CE}$ is the length of the main stream in the catchment. This channel bed area was added to the value of sediment contributing area previously estimated in each basin.

### 2.4.5 Geological index

A geological index $GI$ was calculated on the sediment contributing areas following the methodology proposed by D'Agostino and Marchi (2001). Its value is computed by weighting the score associated to each lithological class (e.g. 5 for Quaternary deposits, 3 for marls and 0.5 for granits) in proportion to the area covered by this lithology in the studied area, here the sediment contributing area. This parameter is a proxy of the relative erodibility of the lithology of the sediment contributing area. The definition of the lithological classes was performed mainly on the basis of national geological maps which account for

superficial formations as fluvial and glacial loose deposits (BD Charm-50 ©BRGM, see https://www.geocatalogue.fr/Detail.do?id=4156). In catchment without mapped sediment contributing area, where even the river channel was too narrow to clearly appear between the mapped vegetation patches (an evidence of weak sediment transport activity), a minimum value of 0.5 was arbitrarily assigned.

### 2.4.6 Index of connectivity IC

Calculation of the Index of connectivity $IC$ was implemented thanks to the SedInConnect stand-alone software (Crema and Cavalli, 2018). It is based on a morphometric algorithm that computes the potential connectivity between hillslopes and a target area from a digital terrain model. Briefly, $IC$ is defined as the logarithm of the ratio between an upslope- and a downslope-component expressing the potential for downwards routing of the sediment produced upslope and the sediment flux path length to the nearest sink along a flow line respectively for each grid-cell of catchment (Borselli et al., 2008; Cavalli et al., 2013). It can be expressed as follows:

$$IC_i = \log_{10} \left( \frac{\overline{W}_i \cdot \overline{S}_i \cdot \sqrt{A_i}}{\sum_i \frac{d_i}{W_i \cdot S_i}} \right) \tag{1}$$

where $i$ is the index of the grid cell at which the computation is performed, $A_i$ is the upslope area (km$^2$), $d$ is the length of the steepest flow line between grid-cell $i$ and the target area (m). $\overline{\chi}_i$ represents the average value of any parameter $\chi$ on the the upslope area of pixel $i$, e.g., of the slope $\overline{S}_i$ (m/m) or of the weighting factor $\overline{W}_i$. $W$ is a weighting factor used to capture the spatial variability of some factor enhancing or damping the sediment transport process. Using the approach of Cavalli et al. (2013), especially suitable for torrents, $W$ is computed from the standard deviation of the residual topography, i.e. difference between the point elevation and the mean average taken on a moving square window of side 5 pixels. A maximum absolute roughness is used to normalize this index. The maximum roughness measured on the Northern French Alps - 75.4 m - was used over all catchments to enable cross-comparison of values between catchments. The computation of the $IC$ considers local obstructions to sediment transfer by providing sink polygons (lakes and glaciers in our analysis).

As output, SedInConnect computes $IC$ values for each grid element. The $IC$ is defined in the range of [-∞, +∞], the result is presented in terms of high or low index, where high values represent a better connectivity to the target. Here, the targets are the catchment outlets. The same DTM than previously was used. Its spatial resolution of 25 m was a bit coarse, 5 m for instance would have been better to capture the typical size of landforms relevant to debris flows and debris floods (Crema et al., 2020; Torresani et al., 2021), but such a detailed DTM was not available at the scale of the study. In addition, the coarse DTM resolution was likely less critical because this study does not address an in-depth analysis of the $IC$ distribution within the catchments but rather seek to extract a lumped variable at the catchment scale.

Indeed, the $IC$ values being variable over the catchment of each debris basin, several statistical values were extracted as potential candidates to be relevant proxies of the catchment-scale connectivity. The mean, median and 95% quantile of the $IC$ ($ICm$, $IC50$ and $IC95$, respectively) were extracted for each catchment. Most of the sediment being supplied by the sediment

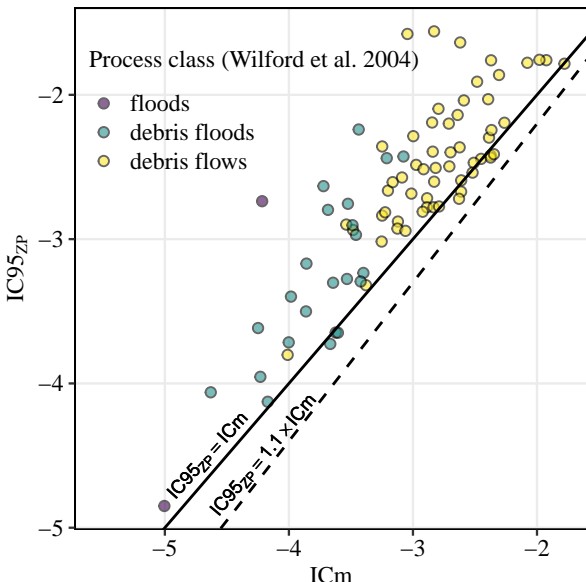

**Figure 6.** Relationship between $IC95_{ZP}$ and $ICm$ of every catchments illustrating that the equation $IC95_{ZP} = 1.1 \cdot ICm$ is a reasonable lower envelope curve.

contributing areas, the median and 95% quantile of the $IC$ were also extracted specifically on the grid elements included in the sediment contributing areas ($IC50_{ZP}$ and $IC95_{ZP}$, respectively).

For basins without delineated sediment contributing area, we assumed that sediment was supplied in a diffuse way throughout the catchment (the channel area was just a proxy of the sediment source area). Since no known source zone was mapped, we assumed the catchment mean IC value to be a relevant proxy of the sediment source IC. A lower envelope curve was defined from a scatter plot of $IC95_{ZP}$ versus $ICm$ to verify this hypothesis on the catchments with known sediment contributing area (Fig. 6). The analysis shows that the ratio of $IC95_{ZP}$ to $ICm$ is generally < 1 except in a few catchments where the ratio is

near 1.1. We arbitrarily assigned $IC95_{ZP} = 1.1 \cdot ICm$ to basins without mapped sediment contributing area. The underlying idea was to assign a value consistent with the distribution of the catchment IC but representative of the lowest connectivity in the data set studied.

     Rather than absolute values of $IC$ that must be interpreted with caution (Cavalli et al., 2013; Heckmann et al., 2018), we tested ratios of $IC$ that could be somewhat normalized. The underlying assumption was that high catchment activity could be

captured by relatively high connectivity of the sediment contributing area to the outlet as compared to the typical connectivity of the catchment. The ratios of the mean and 95% quantile value of $IC$ computed in the sediment contributing area to the values sampled on the whole catchments were also computed ($R_{ICm}$ and $R_{IC95}$, respectively).

## 2.5 Modelling approach

Random forests (RF) and linear regression (LR) techniques were applied to relate geomorphic and climatic characteristics to
sediment yield volumes. Briefly, a RF model comprises an ensemble of individual classification and regression trees (a forest)
from which a final prediction is based on the predictions averaged over all trees (Breiman, 2001). A RF model is created
by drawing several bootstrap samples from the original training data and fitting a single classification tree to each sample.
Independent predictions (i.e. independent of the model fitting procedure) are made for each tree from the observations that
were excluded from the bootstrap sample (the out-of-bag samples OOB). These predictions are aggregated over all trees (the
OOB predictions) and provide an estimate of the predictive performance of the model for new cases. RF models also produce
measures of the importance of each predictor (Liaw and Wiener, 2002; Morel et al., 2020).

Importance analysis helps to select explanatory variables for model formulation. We implemented several prediction models
(both RF and LR) based on one or more explanatory variables. This modelling strategy aims to assess the improvement of
sophisticated models (i.e. multivariate random forests models) compared to parsimonious regressions. Statistical analyses were
performed on R (R Core Team, 2020) and RFs were performed using the package "RandomForest" (Liaw and Wiener, 2002).

To assess the performance of the different models, we performed a leave-one-out (LOO) cross-validation procedure by leaving
out each observation in turn, fitting a model with all remaining data, and then predicting the value of the left-out observation.
This procedure provides an assessment of model performance for undocumented catchments. We quantified model predictive
performance of our LOO predictions using three performance metrics. The coefficient of determination ($R^2$) describes the pro-
portion of the variance in the measured data that is explained by the model with $R^2$=1 for perfect agreement. The percentage
bias (pbias) measures the average tendency of simulated data to be overestimated (pbias<0 %) or underestimated (pbias>0 %)
compared to their observed counterparts. As an error index, we used the RMSE-standard deviation ratio of observations (RSR)
which standardizes the root mean square error (RMSE) using the standard deviation of the observations. Lower RSR values
indicate better model performance, with zero indicating a perfect agreement. See Moriasi et al. (2007) for further details about
calculation and complementarity of these metrics when comparing observed and predicted values.

## 3 Results

### 3.1 Variability of the parameters

The calculated sediment production and associated explanatory variables are visible in Figure 7 and summarised in Table 2.
The whole data set is provided in the Table S1. The distributions and correlations between the variables are shown in the Figure
S1.

In brief, the studied catchments experience specific sediment yield covering three order of magnitude with $V_m/A$, $V_{10}/A$ and
$V_{ref}/A$ ranging from 10 to 13,530 (m³/km²/year), 80 to 42,610 and 150 to 206,000 (m³/km²/event), respectively (Table 2).
The simplistic classification of Wilford et al. (2004) is used throughout this paper to give a crude idea of the type of catchments
(from extended catchments with gentle slopes to very steep, small gullies). The three categories of flood-prone, debris-flood

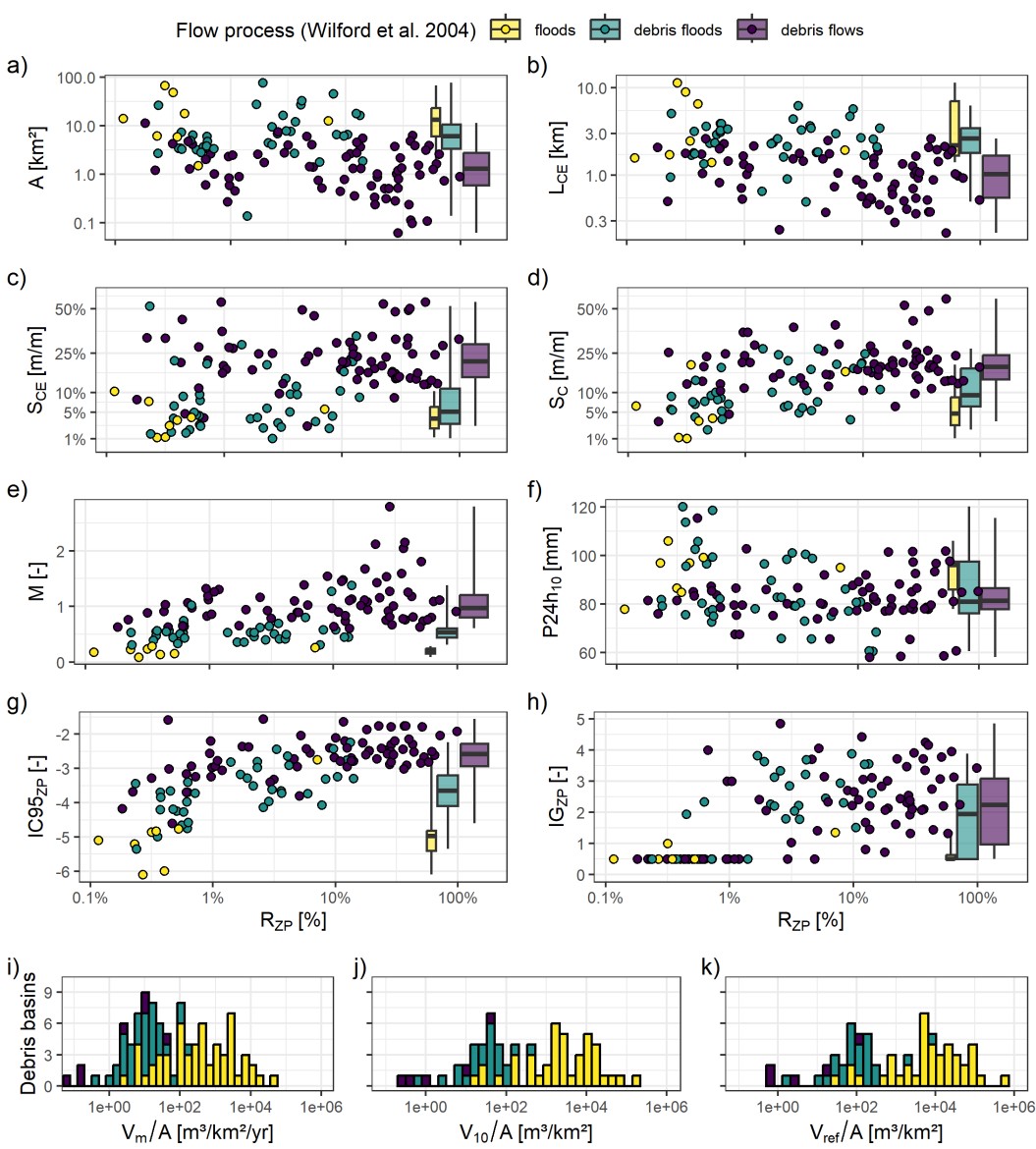

**Figure 7.** Scatter plot of the main calculated variables against the ratio of sediment contributing area $R_{ZP}$: a) catchment area $A$, b) channel length $L_{CE}$, c) channel slope $S_{CE}$, d) fan slope $S_C$, e) Melton index $M$, f) daily precipitation with return period of 10 years $P24h_{10}$, g) quantile 95% of the Connectivity Index extracted in the sediment contributing area $IC95\%_{ZP}$, h) Geological Index of D'Agostino and Marchi (2001) extracted in the sediment contributing area $IG\%_{ZP}$; and histogram of the output variables: i) mean annual specific sediment production $V_m/A$, j) specific event magnitude with a 10 year return period $V_{10}/A$ and k) reference specific event magnitude $V_{ref}/A$.

**Table 2.** Summary of the calculated variables. (-) indicates dimensionless variables.

| Variable | Description | Unit | Min. | Median | Max. |
|---|---|---|---|---|---|
| **Input parameters** | | | | | |
| $P1h_{10}$ | 1 hour rainfall for a 10 years return period | mm | 11 | 21 | 31 |
| $P6h_{10}$ | 6 hour rainfall for a 10 years return period | mm | 32 | 51 | 75 |
| $P24h_{10}$ | 24 hour rainfall for a 10 years return period | mm | 58 | 81 | 120 |
| $A$ | Catchment area | km$^2$ | 0.06 | 2.7 | 77.8 |
| $L_{CE}$ | Stream length | km | 0.22 | 1.7 | 11.5 |
| $S_{CE}$ | Mean channel slope | m/m | 0.01 | 0.16 | 0.54 |
| $S_C$ | Fan slope | m/m | 0.01 | 0.17 | 0.57 |
| $M$ | Melton index | - | 0.09 | 0.77 | 2.8 |
| Class. | Dominant process according to Wilford et al. (2004) | code | 8 BL, 40 DFD, 72 DFW | | |
| $R_{ZP}$ | Ratio of sediment contributing area to catchment area | % | 0.1 | 4 | 98 |
| $IG_{ZP}$ | Geological index of the sediment production areas | - | 0.5 | 3.1 | 5 |
| $ICm$ | Mean value of the connectivity index $IC$ | - | -5.5 | -3.1 | -1.7 |
| $IC50$ | Median value of the IC | - | -5.5 | -3.2 | -1.9 |
| $IC95$ | Quantile 95 of the IC | - | -4.8 | -2.4 | -1.0 |
| $ICm_{ZP}$ | Mean value of the IC in sediment production areas | - | -7.2 | -3.4 | -2.0 |
| $IC50_{ZP}$ | Median value of the IC in sediment production areas | - | -7.2 | -3.4 | -1.9 |
| $IC95_{ZP}$ | Quantile 95 of the IC in sediment production areas | - | -6.0 | -2.9 | -1.5 |
| $R_{ICm}$ | Ratio of $ICm_{ZP}/ICm$ | - | 0.79 | 1.05 | 1.30 |
| $R_{IC95}$ | Ratio of $IC95_{ZP}/IC95$ | - | 0.65 | 1.24 | 1.91 |
| **Output parameters** | | | | | |
| $V_m/A$ | mean annual specific sediment volume | m$^3$/km$^2$/year | 10 | 330 | 13,530 |
| $V_{10}/A$ | 10-years return period of specific sediment volume | m$^3$/km$^2$/event | 80 | 1,450 | 42,610 |
| $V_{ref}/A$ | reference specific sediment volume | m$^3$/km$^2$/event | 150 | 4,180 | 206,000 |

prone and debris-flow prone catchments are however clearly associated to increasing specific catchment production, although some overlapping appear (Figure 7i–k).

This very strong variability of sediment production is directly related to the large variability of the ratio of sediment contributing area to catchment size $R_{ZP}$ (x-axis of Figure 7a–h) ranging between 0.1% and 98% (median 4%). The geological index $GI$ also greatly varies between sediment contributing area fully formed of erosion-sensitive material ($GI = 5$) and in

solid igneous or metamorphic rocks ($GI = 0.5$), with a median $GI = 3.1$ (Figure 7h). It may also be noted that the complete data set consists of generally small catchments (median $A \approx 3$ km$^2$ – Figure 7a), with short stream length (median $L_{CE} = 1.7$ km – Figure 7b), where the slopes are often steep (median $S_{CE} = 0.16$ m/m – Figure 7c–d). The Melton index are relatively high (median $M = 0.77$ – Figure 7e) but values below 0.3, i.e. of basins prone to floods rather than debris floods or debris flow, are observed (Fig. 4). Eight catchments were categorized as flood-prone, 40 as debris flood-prone and 72 as debris flow-prone:

the sample is mostly composed of torrents but not only. Although the absolute value of the connectivity indexes are not easy to interpret, it can be seen that they vary over several units (Figure 7g), e.g. from -6.0 to -1.5 for $IC95_{ZP}$ meaning high variability of the connectivity of the sediment contributing area (recall that $IC$ is a $\log_{10}$ parameter so the upslope to downslope components vary over four to five order of magnitude between the catchments). The 10-year return period rainfall ranges from

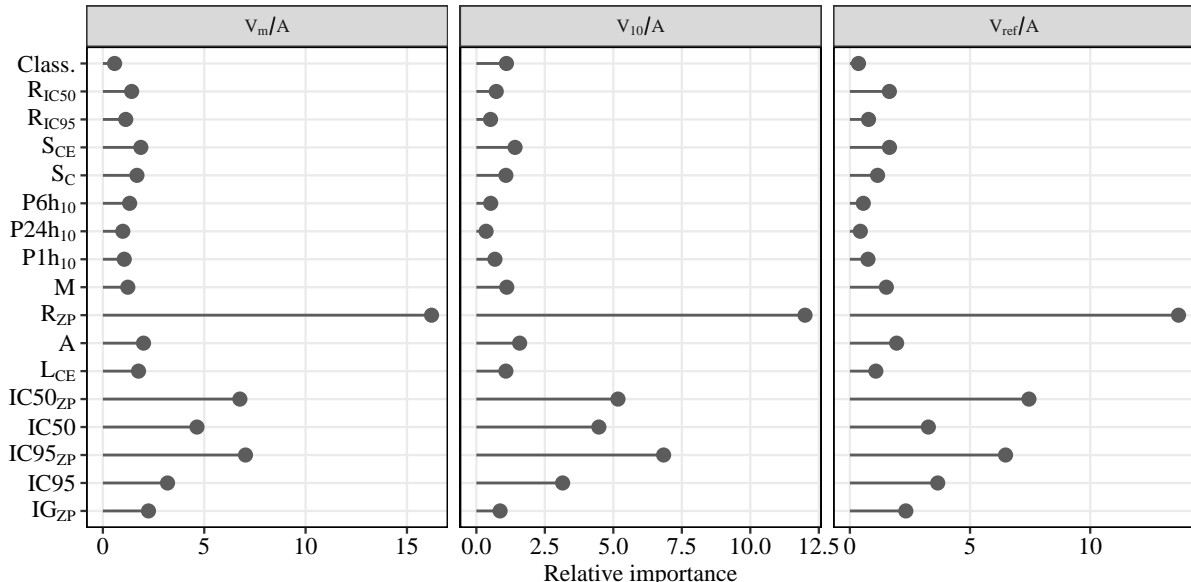

**Figure 8.** Index of importance of the parameters in the prediction of $V_m/A$, $V_{10}/A$ and $V_{ref}/A$ computed by the Random-Forest method.

58 to 120 mm (Figure 7f). In essence, the data set used in this analysis, although focusing on small mountainous catchments located in a temperate climate, comprises a great variability of size, slope, relief, geology, land cover, connectivity and rainfall intensity.

### 3.2 Relative importance of the parameters

The importance of the explanatory variables for predicting the different volumes is shown in Fig. 8. Results showed that the ratio $R_{zp}$ of sediment contributing area was by far the most important predictor of the sediment production volumes. This relationship is presented in Fig. 5. In a lesser extent, the indices of sediment connectivity, especially $IC50_{ZP}$ and $IC95_{ZP}$, have also an influence. Surprisingly, rainfall as well as geomorphic parameters such the Melton index, the fan, the stream slopes or the class of process according to the Wilford et al. (2004) typology showed low importance.

### 3.3 Building of prediction models

Based on the parameter importance, three different model formulations were tested for predicting the volume characteristics $V_m/A$, $V_{10}/A$ and $V_{ref}/A$. The ratio of sediment contributing area $R_{ZP}$ is systematically included as input variable. The first bi-variate equation was defined using also a proxy of the connectivity index, namely $IC95_{ZP}$, which proved to be statistically also an important parameter. Figure 9 shows scatter plots of the sediment production versus $R_{ZP}$ highlighting the strength of this third parameter. Despite its relative low importance identified in Fig. 8, the Melton index was included in the second bi-variate equation due to its simple calculation and common use in the literature. Formulations are shown in Table 3 and

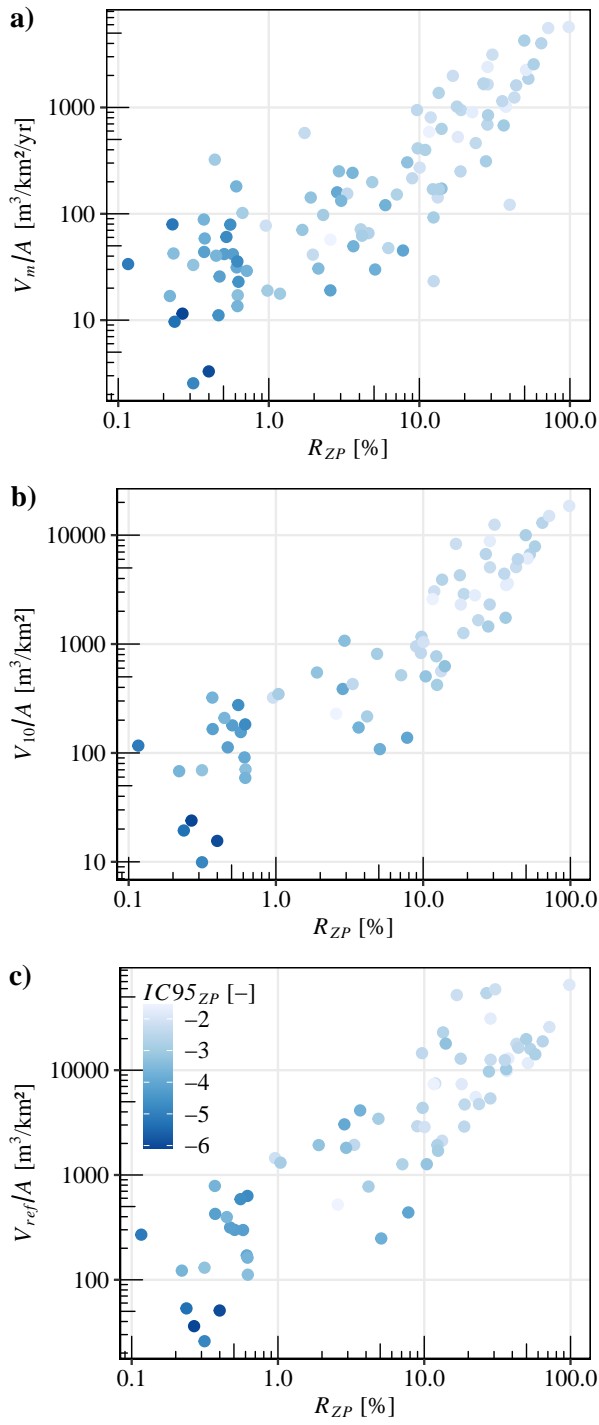

**Figure 9.** Relationships between volumes production variables ($V_m/A$, $V_{10}/A$ and $V_{ref}/A$) and $R_{ZP}$. The color shade varies with $IC95_{ZP}$. It shows that strongly connected sediment contributing area, in light blue, are also preferably detected in catchments with high ratios of production zone $R_{ZP}$.

**Table 3.** Formulation for the different predictive models.

| Model | $V_m/A$ | | $V_{10}/A$ | | $V_{ref}/A$ | |
|---|---|---|---|---|---|---|
| $\#f(R_{ZP})$ | $52 \cdot R_{ZP}^{0.81}$ | (2) | $168 \cdot R_{ZP}^{0.88}$ | (3) | $475 \cdot R_{ZP}^{0.94}$ | (4) |
| $\#f(R_{ZP}, IC95_{ZP})$ | $175 \cdot R_{ZP}^{0.67} \cdot 10^{0.14 \cdot IC95_{ZP}}$ | (5) | $830 \cdot R_{ZP}^{0.70} \cdot 10^{0.18 \cdot IC95_{ZP}}$ | (6) | $1982 \cdot R_{ZP}^{0.77} \cdot 10^{0.16 \cdot IC95_{ZP}}$ | (7) |
| $\#f(R_{ZP}, M)$ | $40 \cdot R_{ZP}^{0.75} \cdot M^{0.17}$ | (8) | $125 \cdot R_{ZP}^{0.81} \cdot 10^{0.2 \cdot M}$ | (9) | $363 \cdot R_{ZP}^{0.87} \cdot 10^{0.19 \cdot M}$ | (10) |

refer to mono-variate $V = f(R_{ZP})$ or bi-variate models $V = f(R_{ZP,M})$ or $V = f(R_{ZP}, IC95_{ZP})$. Other formulations were explored by Morel et al. (2022), but we presented here only the most promising. We tested notably a RF accounting for all parameters and indeed the precision was marginally better than the three simple models we propose. All formulations were implemented using random forests and regressions separately. The $\log$ or $\log_{10}$ of the parameters was used to get distributions closer to Gaussian samples when needed. The equations were then rearranged to get direct estimates of the volumes. That is

why power laws appear in the equations.

### 3.4  Model performances

Summary statistics describing the LOO cross-validation performance of the different methods for the three sediment production volumes $V_m/A$, $V_{10}/A$ and $V_{ref}/A$ are presented in Fig. 10. The comparison of the performances of the different model formulations shows in general comparable performances between the RF and LR models. Contrary to what was expected, the

RF models do not bring improvements compared to the LR models. For example, for $V_{10}/A$ models, R2 values range from 0.77 to 0.81 and from 0.77 to 0.80 for models using RF and LR, respectively. The errors are also smaller for the models using random forests (RSR indicator).

For all models, the percentage bias is negligible (on the order of 1 %). The performances are also evaluated in Figure 10 in terms of the proportions of the ratio of predicted divided by observed absolute values to be in the intervals $[1/2; 2]$ and $[1/5; 5]$.

We notice that about 30 % of the predictions fall in the first interval, and about 50 % in the second. This gives a sense of the precision of such equations: they capture a relevant first approximation but cannot be very precise. Interestingly, bi-variate equations are only marginally more precise than the mono-variate equations, both using the simple Melton index $M$ or the more sophisticated connectivity index $IC95_{ZP}$. Plots of the observed and predicted values, as in Fig. 11, are provided for each formulation in the Figures S2, S3 and S4.

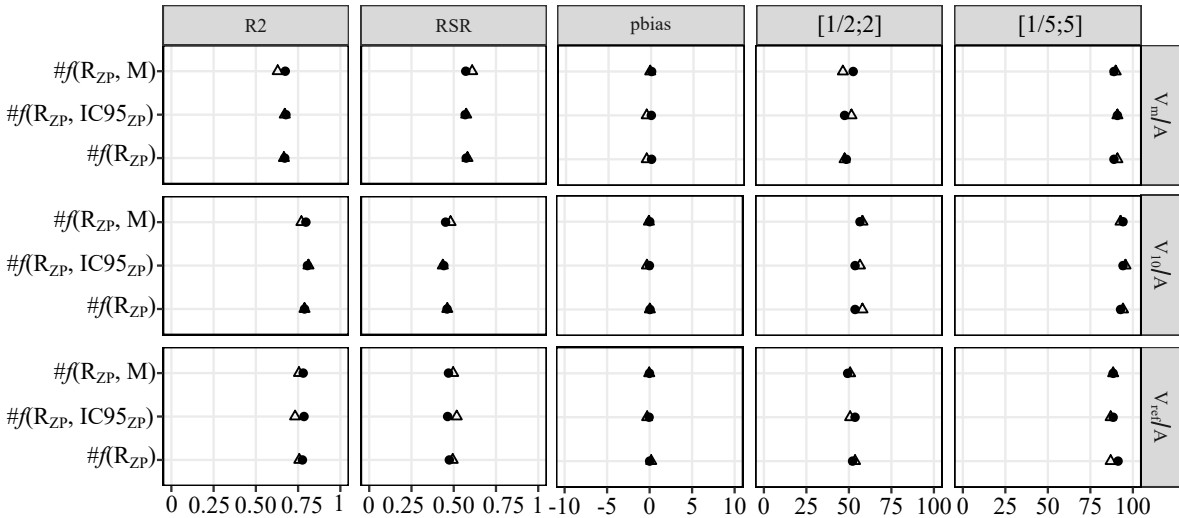

**Figure 10.** Efficiency criteria values obtained for leave-one-out cross validations of random forest and regressions model results. R2, RSR and pbias were evaluated on the specific volumes (i.e. $V_m/A$, $V_{10}/A$ and $V_{ref}/A$). The analysis of the proportion of the predicted values to be included in the reference intervals were evaluated on the absolute volumes (i.e. $V_m$, $V_{10}$ and $V_{ref}$).

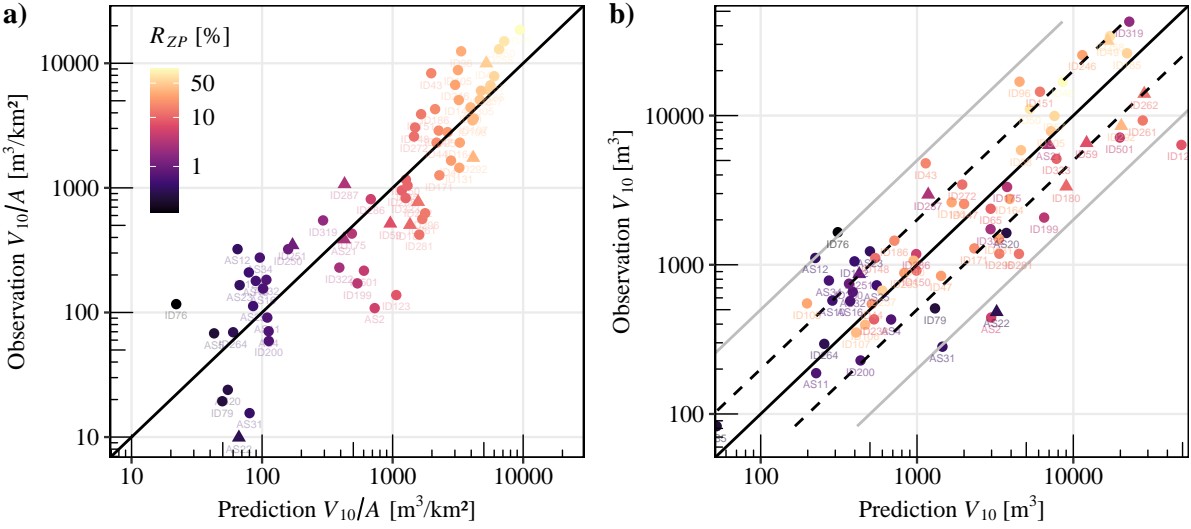

**Figure 11.** Exemple of observed against LOO predicted values for a) the specific volume $V_{10}/A$ and b) the the absolute volume $V_{10}$ (application of Equation 3). The black lines show perfect agreement between observed and predicted (i.e. $1:1$). The dotted line and the gray lines show, respectively, reference intervals $[1/2;2]$ and $[1/5;5]$. The code name of each catchment as provided in the full data set in the supplement is displayed near the dots.

## 4 Discussion

### 4.1 Input parameters

The key role of the sediment contributing area on the solid production of torrents was for instance stressed and studied by D'Agostino and Bertoldi (2014) who suggest an approach to prioritize the treatment of the various active areas. Alternatively, the present paper seeks to predict the sediment production volumes, and provides an original dimension by directly including this sediment contributing area, rather than the full catchment area, in the equations. Conceptually, this parameter captures most of the potential sediment supply of the catchment since sediment production is much higher on bare soils than under forest and vegetation cover (Cerda, 1999; Vanacker et al., 2007; de Vente et al., 2011; Mishra et al., 2019; Carriere et al., 2020). The strong correlation between sediment yield and erosive area has already been observed in the analyses conducted on a smaller data set comprising the most active catchments of the same region by Peteuil and Liébault (2011) regarding mean annual and event sediment production. Altmann et al. (2021) also proposed an automatic DEM analysis based on slopes and distance to drainage network to predict the sediment contributing area, which was well correlated to the mean annual sediment production. However, as shown in Table 1, this sediment contributing area did not appear until now in the other empirical equations proposed in the literature to predict event magnitude in mountain torrents.

The strong correlation with $R_{ZP}$ shows that predictive models are well adapted to catchments where sediments mainly come from surface erosion (rill, sheet or gully erosion) but may significantly underestimate volumes in catchments where a significant proportion of inputs are from other, less visible sources (e.g. forested or vegetated landslides). If the sediment sources of a catchment are of this type, our methods should be dismissed and a method which does not include an estimated sediment contributing area should be preferred. Using maps of landsliding areas as sediment contributing area could be an idea to test our method on such catchments but we suspect that the elementary sediment production, in term of m$^3$/km$^2$ of active area, is likely much higher for landslides than for gullies, bare soil and cliffs (Rickenmann and Koschni, 2010). This would be an interesting research question to explore.

The sediment production volumes did not exhibit strong relationships with the geological index. These results are consistent with the observation of Marchi and D'Agostino (2004) who showed the same result on their sample. It is also possible that this variable does not capture the complex interactions of geological, geomorphological and climatic conditions on sediment production. It must also be acknowledged that in most mountains of France, as well as, for instance, in Italy, Switzerland or Austria, torrent control works and reforestation master-plans were implemented in the past and a strong spontaneous reforestation occurred due to rural depopulation (Piton et al., 2017). Hence, the sediment contributing areas studied in this paper, i.e. areas where bare soil remains nowadays, are necessarily very erosive and active areas. The other areas that used to be active but that had geological features slightly less prone to erosion are most likely now stabilized and vegetated. It is also interesting to mention that we tested an extraction of the geological index at the scale of the catchments, rather than only on the sediment contributing areas. It proved useless and uncorrelated to the sediment production: many catchments have for instance a geology mostly composed of moraine, supposedly a lithology prone to erosion, but are fully vegetated by mature forest and thus do not produce much sediment.

The sediment production volume were also poorly correlated to the rainfall proxies. We assume that the temperate climate and associated meteorology of the studied region (Figure 1b), does not vary enough to capture this possible effect and eventual feedback loop on the vegetation cover.

The results show that morphometric variables ($M$, $S_{CE}$, $S_C$) have a limited influence on sediment production. Precise topographic data were seldom available for the studied catchments and we made the choice to calculate the morphometric variables using a 25 m resolution DTM. This resolution is quite coarse, especially in the case of small steep watersheds. For a few catchments where more accurate DTM were available, we cross-compared these precise surveys with the values of slopes extracted in our study. The values were reasonably accurate for reaches flowing on large alluvial fans, well coupled with the upstream active basin (Harvey, 2002). Conversely, in narrow valleys or if the torrents were incised in the fans, the coarse DTM did not capture the actual channel profile which had a characteristic scale lower than the DTM resolution: incised channel bed and drops related to check dams were for instanced smoothed. In such case, the slopes were not accurately estimated. Repeating this analysis using more precise DTM would be interesting when such data will be available at a regional scale.

The classification of the dominant process type according to Wilford et al. (2004) also does not appear as a meaningful variable in our analysis. This is surprising because typical event magnitude of debris flows are usually quite higher than of bedload events for a given catchment size (Rudolf-Miklau and Suda, 2013; Hübl, 2018). Rather than relying on a simplistic classification as we did here due to a lack of information, in further research it could be of interest to classify more precisely the type of process involved in each catchment, and, if possible, for each events based on field and historical evidences (D'Agostino, 2013; Kaitna and Hübl, 2012). Then it would be possible to fit extreme values predictions that would be process-specific in addition to be catchment-specific. Extending the dataset to other sites and eventual regions would be necessary not to perform such analyses on excessively small sub-samples.

Another contribution of this work is the inclusion of a connectivity index, namely the quantile 95% of the $IC$ extracted on the sediment contributing area, into the empirical equation. The analysis of the importance of the various explaining factor produced by the random forest analysis shows that the proxies associated to the $IC$ are all of higher importance than any other variable except for the ratio of sediment contributing area $R_{ZP}$ (Fig. 8). Indicators dedicated to the analysis of the sediment connectivity are numerous. As pointed out by the review of Heckmann et al. (2018), these indices, despite their high interest, are seldom used as input tool for simple predictive models of sediment delivery by catchment. Mapping $IC$ is now simple and fast with openly accessible tools (Crema and Cavalli, 2018; Martini et al., 2022). Using a DTM with a relevant resolution (Crema et al., 2020), such distributed information can be useful to shed a new light on the catchment morphology. With the Eqs. (5), (6) and (7), such indices are also reused to link this catchment features with their associated sediment export.

## 4.2 Discussion about the predictive models

The predictive models developed in this work provide an approximate assessment of sediment yield, and their use should be restricted to the physical context (geological, geomorphological, climatic) where they have been developed. However, the diversity of the torrents studied makes it worth testing their application in other regions. Including fully vegetated catchments with very small sediment production in our data set, and not only the most active torrents, is also a strength of our work: most

mountain catchments in the Northern Alps, and in many other mountain ranges, are vegetated and poorly active. Nonetheless, some assets are located on their alluvial fans and practitioners need methods to study and predict their infrequent solid production. We believe that this contribution will help to address such weakly active hydrosystems with very few sediment sources.

The performance analysis shows that these equations provide at most a relevant order of magnitude of sediment production. In this paper, we reviewed some existing empirical equations used to estimate catchment solid production (Tab. 1) and propose new equations, basically three equations for each variable to predict (mean annual production or event magnitude). We are regularly asked by practitioners which equation or method should be used. In practice, we apply all equations of Tab. 1 as well as the three new equations developed here. We know that none of them is very precise. Many underlying processes make the actual sediment production of mountain catchment very complex to predict. We do not think that using an equation with many input parameters would lead to a serious gain in precision: we tested a random forest model accounting for all parameters and indeed the precision was marginally better than the three simple models we propose. Each equation captures a given trend. Collectively, these equations constitute a body of knowledge helping to bound the behaviour of mountain catchments. Using several equations rather than one provide multiple estimations. This also highlights the lack of precision of these equations. Using one single equation could give a false sense of precision to inexperienced users. By using them regularly, and confronting them to other sources of information, users learn about the bias and behaviour of each equation or group of equations guiding them in one of the many educated guesses that any debris-flow or debris flood hazard assessment involves.

Such empirical equations are obviously only one type of tool between many others. Debris-flow and debris flood hazard assessments require further *in situ* and historical analyses adapted to the stage of study (Jakob, 2021). When possible, the practice in France (also consistent with Jakob et al., 2022), is to compare the results of empirical equations with: (i) in-depth historical analysis (Marchi and Cavalli, 2007; D'Agostino, 2013). Such analyses sometimes enable to gather sufficient information to perform a local extreme value fit as in this work. Most of the time they only give an order of magnitude of one or a few extreme events which is still an interesting information. (ii) Simple computations can be done using rainfall data associated with hypothesis on runoff coefficient and on solid concentration of the flows (e.g. Marchi and D'Agostino, 2004; Rickenmann and Koschni, 2010). (iii) Field visits finally help to map potential debris sources in term of length of active gullies or erodible bed and associated possible erosion rate in $m^3$/m of channel (e.g. Hungr et al., 1984; Marchi and D'Agostino, 2004). The latter exercise is key to ensure that there is indeed available material to form debris flows and debris floods and helps correcting other empirical approaches, for instance in catchments with extended bare rock area of strong igneous rock that are often supply limited.

## 5   Conclusion

Using a unique data set of sediment dredging in about one hundred debris basins and associated historical information, we estimated the mean annual and event-driven sediment production of torrents located in the Northern French Alps. Several geomorphic indicators of their catchments were extracted including the catchment area, the sediment contributing area, the

Melton index, as well as several statistical values of the index of connectivity IC computed on these catchments and other relief, lithological and rainfall indexes. We used these geomorphic parameters to try to predict the sediment supply of these catchments using several statistical methods. Results showed that the ratio of connected eroding areas was the most important predictor of the sediment production volumes.

In line with the previous works synthesized in Table 1, this paper demonstrates that simple equations can predict sediment yield in torrent catchments with independent basin parameters that are simple and easy to determine. These simple equations are much faster and easier to use than sophisticated models as random forest methods, that surprisingly, did not lead to a better accuracy. Despite their limitations, predictive equations in Table 3 provide refined estimation methods of sediment production volumes. Because our models were built using a wide range of torrent types, they have the potential to be tested and applied in

different regions.

These models complete the existing body of empirical methods that are used to assess debris-flow and debris flood hazard as well as to design protection measures. By including explicit mapping of the sediment contributing area and quantiles of the index of connectivity, our methods push the practitioners to focus on sediment sources and to analyse the sediment connectivity of the catchments: such analyses are not only useful to fuel our equations but will shed new light on the studied sites as

compared to the classical analysis of slopes, relief and catchment size.

*Code and data availability.* Many more details on this work are available in the research report of Morel et al. (2022) available here https://hal.archives-ouvertes.fr/hal-03549827. The parameters describing each catchment (rainfall, morphology and sediment production) are available in a synthetic table in Table S1, as well as the time series of sediment production of each catchment and the associated extreme value fit in Figure S5. Maps of every catchments and the raster grid of the weighting factor $W$ usable to apply our method on the whole

French northern alps are available at the following repository: https://entrepot.recherche.data.gouv.fr/dataverse/HydroDemo. The R codes used to perform the analysis are available upon reasonable requests by directly contacting the first or second authors.

## Appendix: Nomenclature

$\chi_{...}$   Generic parameter defined locally in the text

A   Catchment area [km$^2$]

GI   Geological Index of D'Agostino et al. (1996) [-]

IC...   Quantile of probability ...% of the Connectivity Index [-]

IC...$_{ZP}$   Quantile of probability ...% of the Connectivity Index extracted only on the sediment contributing area [-]

L   Length of the main river channel [km]

M   Melton index: catchment relief / $\sqrt{A}$ [-]

$R_{ZP}$     Ratio of sediment contributing area to catchment area [-]

$S_C$      Slope of the alluvial fan [m/m]

S or $S_{CE}$  Slope of the channel controlling the sediment transport [m/m]

T        Return period [year]

V        Solid volume associated to an event [m$^3$]

$V_m$      Mean annual solid volume [m$^3$]

$V_{...\%}$    Quantile ...% of a solid transport sample [m$^3$]

$V_{ref}$    Solid volume with return period of 100 yrs or largest sediment volume observed if higher [m$^3$]

$V_T$      Solid volume with return period of T yrs [m$^3$]

*Author contributions.* MM performed the statistical analysis with advises from GE and wrote the first version of the paper. GP and CLB
supervised the work. GE supervised the statistical analysis. All authors helped finalizing the paper.

*Competing interests.* The authors declare no conflict of interest.

*Acknowledgements.* This study is part of the HYDRODEMO project which is financed by the European Union through the FEDER-POIA
program and by state funds through the FNADT-CIMA program. The authors would like to thank the French torrent control service (ONF-
RTM) and the many catchment stakeholders who provided dredging data on the torrents. Many thanks to Alexandre Mas for providing rain
data COMEPHORE on the studied catchments. The authors also acknowledge useful and relevant comments by Lorenzo Marchi and an
anonymous referee that helped much to improve the manuscrit.

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
