# Peer review of "Statistical Modelling of Sediment Supply in Torrent Catchments of the Northern French Alps"

_EGUsphere, 2022_

## Referee Comment (RC1)

**Reviewer's comment on egusphere-2022-1494**

**General comment**

This study focuses on the assessment of coarse sediment volumes in mountain catchments, which is an important issue for the control of sediment-related hazards and risks. The developmjent of statistical models that relate sediment volumes to catchment characteristics is not novel, but this work shows some valuable features, namely the frequency analysis of time series of sediment volume data, the inclusion of sediment connectivity among the predictive factors, and a careful evaluation of model performances. The analysis is based on a very good dataset and is performed using up-to-date statistical techniques.

I am reporting below some comments hoping that they could be useful in the paper's revision.

**Table 1.**

The equations (b) and (c) of Rickenmmann (1997) define envelope curves (EC).

The equations involving catchment area, slope and the geological index, proposed by D'Agostino et al. (1996) and D'Agostino and Marchi (2001) could be removed because they have a similar structure to the equation by Marchi and D'Agostino (2004), which is based on a larger dataset.

The authors could consider the equations proposed by Marchi et al. (2019) that link debris-flow volume ($V_{DF}$) to catchment area ($A_B$) for various quantiles. These equations, which are based on a sample of 809 debris-flow volumes in the Eastern Italian Alps (https://doi.pangaea.de/10.1594/PANGAEA.896595), are not intended as predictive tools, while they aim at defining the scaling relationships between these variables. We observe, however, that the equation for the 99 percentile ($V_{DF} = 77000 \pm 7000 \cdot A_B^{(1.01 \pm 0.06)}$) is similar to the empirical envelope line (fitted by eye) proposed for the same region by D'Agostino and Marchi (2001).

**Type of flow process**

The records of sediment transport events include information on the type of flow process, i.e. debris flow of flood (line 84). It is likely - and some details about that would be welcome - that some catchments were affected by only one type of sediment transport process, whereas other catchments experienced both floods with intense bedload and debris flows. It seems to me that the information on the type of sediment transport has not been fully exploited, and events with different transport mechanisms have been processed together to derive the equations for predicting sediment volumes. Developing separate equations for floods and debris flows could have permitted to gain significant insights into the capability of different processes in delivering sediment in the catchments of the studied region. I understand, however, that the smaller sample size for separate processes could have been detrimental to the robustness of the analysis.

The issue of the type of sediment transport processes also arises in section 2.3.3 with the application of two approaches based on catchment topography to the recognition of the dominant sediment transport. Little is said about the agreement between the transport process predicted according to Wilford et al. (2004) and the transport process documented from debris basin dredging and RTM archives (section 2.1).

I presume that the three classes of sediment transport processes considered in Figure 5 are based on the application of the approach by Wilford et al. (2004) (i.e., not on archive data): this could be clearly stated in the caption. I suggest describing the three classes in Figure 5 using a legend within the figure instead of the caption.

**Sediment contributing areas**

Does the recognition of "bare soil" permits discriminating bare soil from bare (outcropping) rocks? Both bare soil and bare rocks, if connected to the channel network, supply sediment for debris flows and fluvial transport. However, the erosion rate is usually much higher on bare soil/debris than on outcropping rock.

Regarding the use of channel area as a proxy of sediment source area (lines 165-166, 207), it could be of some interest to remember that the area of main stream channel was found to be significantly correlated to the sedimentation of reservoirs in one of the earliest studies that applied multiple regression to sediment yield estimation (Anderson, 1949).

**Roughness in sediment connectivity**

The rather coarse DTM resolution could hamper the computation of the topographic roughness as an index of impedance to sediment transfer across the landscape. The optimal DTM resolution in the computation of the topographic roughness depends on the spatial scale of the geomorphic processes investigated. This issue is discussed in Crema et al. (2020) and a conversation on GitHub: https://github.com/HydrogeomorphologyTools/Connectivity-Index-ArcGIS-toolbox/issues/4

However, in the wide frame of this study, which computes the index of connectivity IC to derive an independent variable lumped at the catchment scale, this detail on the computation of the topographic roughness can be considered less critical than for studies aimed at representing sediment connectivity in a distributed way.

**Discussion**

The authors frankly acknowledge the limits in the accuracy of the developed equations, which "capture a relevant first approximation but cannot be very precise" (line 290). The suitability of statistical equations for sediment volumes prediction has received different opinions in the literature. In the case of debris flows, Rickenmann (1999) found that some predictive equations "may overestimate the actual debris-flow volume by up to a factor of 100" and recommended "to make a geomorphologic assessment of the sediment potential rather than using these equations". This statement could sound too drastic, especially if the equations are based, like in this study, on careful data collection and thorough statistical analysis. However, in my opinion, the geomorphological assessment of mobilizable debris remains the core of any estimation of sediment volumes in torrent catchments, while the statistical equations relating sediment volumes to catchment parameters provide at most a useful comparison with the geomorphological estimates. I don't know if the authors agree with my point of view: I am proposing it as a hint for extending the discussion on the application of the predictive models developed in this study, including the integration with other methods, now briefly mentioned in lines 370-371.

**References**

Anderson, H. W.: Flood frequencies and sedimentation from forest watersheds. Eos, Transactions American Geophysical Union, 30(4), 567-586, 1949.

Crema, S., Llena, M., Calsamiglia, A., Estrany, J., Marchi, L., Vericat, D., Cavalli, M.: Can inpainting improve digital terrain analysis? Comparing techniques for void filling, surface reconstruction and geomorphometric analyses. Earth Surface Processes and Landforms, 45(3), 736-755, doi: 10.1002/esp.4739, 2020.

D'Agostino, V. and Marchi, L.: Debris flow magnitude in the Eastern Italian Alps: data collection and analysis, Physics and Chemistry of the Earth, Part C: Solar, Terrestrial & Planetary Science, 26, 657–663, 2001.

Marchi, L. and D'Agostino, V.: Estimation of debris-flow magnitude in the Eastern Italian Alps, Earth Surface Processes and Landforms, 29, 207–220, https://doi.org/10.1002/esp.1027, 2004.

Marchi, L., Brunetti, M.T., Cavalli, M., Crema, S.: Debris-flow volumes in northeastern Italy: relationship with drainage area and size probability. Earth Surface Processes and Landforms, 44(4), 933-943, doi: 10.1002/esp.4546, 2019.

Rickenmann, D.: Empirical relationships for debris flows. Natural Hazards 19, 47-77, 1999.

Wilford, D. J., Sakals, M. E., Innes, J. L., Sidle, R. C., and Bergerud,W. A.: Recognition of debris flow, debris flood and flood hazard through watershed morphometrics, Landslides, 1, 61–66, https://doi.org/10.1007/s10346-003-0002-0, 2004.

---

## Author Comment (AC1)

Responses to the comments of Dr Lorenzo MARCHI on egusphere-2022-1494

**Editorial template**

Each comments of the reviewer are presented in the normal font.

> *The responses of the authors are indented and written in italic.*

**General comment**

This study focuses on the assessment of coarse sediment volumes in mountain catchments, which is an important issue for the control of sediment-related hazards and risks. The development of statistical models that relate sediment volumes to catchment characteristics is not novel, but this work shows some valuable features, namely the frequency analysis of time series of sediment volume data, the inclusion of sediment connectivity among the predictive factors, and a careful evaluation of model performances. The analysis is based on a very good dataset and is performed using up-to-date statistical techniques.

I am reporting below some comments hoping that they could be useful in the paper's revision.

> *We thank very much Dr Marchi for this feedback. We feel lucky to benefit for his comments and long experience on this topic.*

**Table 1.**

The equations (b) and (c) of Rickenmmann (1997) define envelope curves (EC).

> *Thanks, we were not sure. This will be added.*

The equations involving catchment area, slope and the geological index, proposed by D'Agostino et al. (1996) and D'Agostino and Marchi (2001) could be removed because they have a similar structure to the equation by Marchi and D'Agostino (2004), which is based on a larger dataset.

> *Thanks for this information. The equations of 1996 will be removed. We however would like to keep the mention to the paper of 2004 to recall the simple envelope curve V = 70 000 A to echo with the paper of 2019 mentioned below. Also the idea to add the return period into the envelop curve of the last equation taken from D'Agostino and Marchi 2001 deserves being mentioned.*
>
> *We will add the remark that the sample of 2001 was part of the one of 2004 on the line describing the latter.*

The authors could consider the equations proposed by Marchi et al. (2019) that link debris-flow volume ($V_{DF}$) to catchment area ($A_B$) for various quantiles. These equations, which are based on a sample of 809 debris-flow volumes in the Eastern Italian Alps (https://doi.pangaea.de/10.1594/PANGAEA.896595), are not intended as predictive tools, while they aim at defining the scaling relationships between these variables. We observe, however, that the equation for the 99 percentile ($V_{DF}=77000\pm7000\cdot A^{(1.01\pm0.06)}$) is similar to the empirical envelope line (fitted by eye) proposed for the same region by D'Agostino and Marchi (2001).

*Very good point: we will add the reference and the equation in Table 1 and will add as a remark in the introduction (L36):*

> *"Interesting trends can nonetheless be capture on small samples: the envelope curve V = 70, 000 · A that was eye-fitted by D'Agostino and Marchi (2001) on 84 events is for instance very close from the quantile equation $V_{99\%} = 77, 000 · A^{1.01}$ proposed by Marchi et al. (2019) for the same region on a ten time larger dataset."*

**Type of flow process**

The records of sediment transport events include information on the type of flow process, i.e. debris flow of flood (line 84). It is likely - and some details about that would be welcome - that some catchments were affected by only one type of sediment transport process, whereas other catchments experienced both floods with intense bedload and debris flows. It seems to me that the information on the type of sediment transport has not been fully exploited, and events with different transport mechanisms have been processed together to derive the equations for predicting sediment volumes. Developing separate equations for floods and debris flows could have permitted to gain significant insights into the capability of different processes in delivering sediment in the catchments of the studied region. I understand, however, that the smaller sample size for separate processes could have been detrimental to the robustness of the analysis.

The issue of the type of sediment transport processes also arises in section 2.3.3 with the application of two approaches based on catchment topography to the recognition of the dominant sediment transport. Little is said about the agreement between the transport process predicted according to Wilford et al. (2004) and the transport process documented from debris basin dredging and RTM archives (section 2.1).

> *This is also a key point. Dr Marchi is absolutely correct, our sample is composed of a mixture of bedload-prone and debris flow-prone basins, as well as basins experimenting mixed regimes with routine events being often bedload and extreme events sometimes being debris flows. This is a very complicated question.*
>
> *Actually, the RTM database mentioned L84 is sadly not systematically clear about the process type and often does not cover small events that nonetheless resulted in a dredging of the basin (an event is recorded only if it triggered some damage). This point will be clarified in the revised version:*
>
> > *"Briefly, this database provides information on past events that triggered damages in the catchments (to protection structures, roads or buildings), giving details of any causes, eventually information on the process type (i.e. debris flow or flood) and sometime also the volumes of sediment transported."*
>
> *Most dredging data being for not extreme events, we consequently have too often poor knowledge or unclear evidences of the process type of the event. In several catchments, we do not even know the main dominant process. This would require interviews of the catchment managers of the one hundred basins and careful cross control by field visit. This is out sadly of the scope of this work. Therefore, we did not exploited this information*

*mostly because it is missing (and we agree that it would be of great interest).*

*Another elements: when the processes are clearly identified as simple bedload or clear debris flows, it indeed usually make sense to analyse them separately: the unit volume of the latter being generally much higher. To our experience it becomes more complicated when debris floods events are found and more generally for catchments where several, mixed processes occur and change along the propagation. We will provide the following complementary information at the end of §2.3.3 (near L.148):*

> *"For these reasons, we adopted the method of Wilford et al. (2004) for the study. Only this automatic classification was used without exhaustive cross-checking with field evidences due to the lack of availability of relevant and rigorous documentation on this question. In addition, many catchments experience mixed regimes where frequent and small events are rather related to bedload transport while infrequent, larger events might be debris flows (e.g. Theule et al., 2012; Marchi and Cavalli, 2007; Hübl, 2018): assigning a category is thus challenging. We decided to use the simple classification proposed by Wilford et al. (2004) – which is straightforward to use even on a undocumented catchment – simply to test if these classes emerged as sub-samples having clearly different sediment production capacities. It must be acknowledge that this is only a simplistic indicator and not a field-based evidence of a flow process type."*

*Finaly: the whole sample is only of about a hundred catchment, i.e. not very large. Thus, as pointed by Dr Marchi in his comment, splitting it in two or three subsamples whose boundaries would have somewhat been arbitrary due to a lack of information would have very probably decrease the statistical rigour of the analysis. This point was missing and deserved to be addressed as pointed by Dr March. We thus added to the discussion on the input parameters (§4.1) a paragraph on this idea:*

> *"The classification of the dominant process type according to Wilford et al. (2004) also does not appear as a meaningful variable in our analysis. This could appear surprising because typical event magnitude of debris flows are usually quite higher than of bedload events for a given catchment size (Rudolf-Miklau and Suda, 2013; Hübl, 2018). Rather than relying on a simplistic classification as we did here due to a lack of information, in further research it could be of interest to classify more precisely the type of process involved in each catchment, and, if possible, for each events based on field and historical evidences (D'Agostino, 2013; Kaitna and Hübl, 2012). Then it would be possible to fit extreme values predictions that would be process-specific in addition to be catchment-specific. Extending the dataset to other sites and eventual regions would be necessary not to perform such analyses on excessively small sub-samples."*

I presume that the three classes of sediment transport processes considered in Figure 5 are based on the application of the approach by Wilford et al. (2004) (i.e., not on archive data): this could be clearly stated in the caption. I suggest describing the three classes in Figure 5 using a legend within the figure instead of the caption.

*A legend will be added to the figure and its title will state that indeed, the classes are taken from Wilford et al. 2004. See below. Thanks for the suggestion.*

[Figure]

*Figure 5. Relationship between IC95$_{ZP}$ and ICm of every catchments illustrating that the equation IC95$_{ZP}$ = 1.1 · ICm is a reasonable lower envelope curve.*

**Sediment contributing areas**

Does the recognition of "bare soil" permits discriminating bare soil from bare (outcropping) rocks? Both bare soil and bare rocks, if connected to the channel network, supply sediment for debris flows and fluvial transport. However, the erosion rate is usually much higher on bare soil/debris than on outcropping rock.

*Indeed, this is a shortcut that we miss to mention. We pooled all surfaces of connected bare soil and rock in the sediment contributing area. The difference in lithology and associated variable erosion rate was supposed to be analysed in the Geological Index (but did not prove statistically meaningful). We will add in the section describing the sediment contributing area:*

*"It is worth mentioning that bare bedrock is also included in the sediment contributing area in our approach. Although bedrock also produce sediment, bare soil has usually a higher to much higher erosion rate. Any surface area of connected bare soil or rock is however considered equally in our approach, their lithological*

> *differences is assessed in the Geological Index presented in the next sub-section."*

Regarding the use of channel area as a proxy of sediment source area (lines 165-166, 207), it could be of some interest to remember that the area of main stream channel was found to be significantly correlated to the sedimentation of reservoirs in one of the earliest studies that applied multiple regression to sediment yield estimation (Anderson, 1949).

> *Thank you very much for this reference that we did not know. It will be added.*

**Roughness in sediment connectivity**

The rather coarse DTM resolution could hamper the computation of the topographic roughness as an index of impedance to sediment transfer across the landscape. The optimal DTM resolution in the computation of the topographic roughness depends on the spatial scale of the geomorphic processes investigated. This issue is discussed in Crema et al. (2020) and a conversation on GitHub: https://github.com/HydrogeomorphologyTools/Connectivity-Index-ArcGIS-toolbox/issues/4
However, in the wide frame of this study, which computes the index of connectivity IC to derive an independent variable lumped at the catchment scale, this detail on the computation of the topographic roughness can be considered less critical than for studies aimed at representing sediment connectivity in a distributed way.

> *Very relevant remark: indeed, it is because we extracted with the same approach and normalization method the IC on our dataset, and, most of all, because we extracted just a lumped value that we allow ourselves to use such a coarse DTM. We will add the following remark:*
>
> > *"In addition, the coarse DTM resolution was likely less critical because this study does not address an in-depth analysis of the IC distribution within the catchments but rather seek to extract a lumped variable at the catchment scale."*

**Discussion**

The authors frankly acknowledge the limits in the accuracy of the developed equations, which "capture a relevant first approximation but cannot be very precise" (line 290). The suitability of statistical equations for sediment volumes prediction has received different opinions in the literature. In the case of debris flows, Rickenmann (1999) found that some predictive equations "may overestimate the actual debris-flow volume by up to a factor of 100" and recommended "to make a geomorphologic assessment of the sediment potential rather than using these equations". This statement could sound too drastic, especially if the equations are based, like in this study, on careful data collection and thorough statistical analysis.
However, in my opinion, the geomorphological assessment of mobilizable debris remains the core of any estimation of sediment volumes in torrent catchments, while the statistical equations relating sediment volumes to catchment parameters provide at most a useful comparison with the geomorphological estimates. I don't know if the authors agree with my point of view: I am proposing it as a hint for extending the discussion on the application of the predictive models developed in this study, including the integration with other methods, now briefly mentioned in lines 370-371.

*We fully agree with Dr Marchi on this point. The end of the discussion was indeed too short. We will add to the end of the discussion a few more details:*

*"Such empirical equations are obviously only one type of tool between many others. Debris-flow and debris flood hazard assessments require further in situ and historical analyses adapted to the stage of study (Jakob, 2021). When possible, the practice in France (also consistent with Jakob et al., 2022), is to compare the results of empirical equations with: (i) in-depth historical analysis (Marchi and Cavalli, 2007; D'Agostino, 2013). Such analyses sometimes enable to gather sufficient information to perform a local extreme value fit as in this work, most of the time they only give an order of magnitude of one or a few extreme events. (ii) Simple computations can be done using rainfall data associated with hypothesis on runoff coefficient and on solid concentration of the flows (e.g. Marchi and D'Agostino, 2004; Rickenmann and Koschni, 2010). (iii) Field visits finally help to map potential debris sources in term of length of active gullies or erodible bed and associated possible erosion rate in m3/m of channel (e.g. Hungr et al., 1984; Marchi and D'Agostino, 2004). The latter exercise is key to ensure that there is indeed available material to form debris flows and debris floods and helps correcting other empirical approaches, for instance in catchments with extended bare rock area of strong igneous rock that are often supply limited."*

**References**

Anderson, H. W.: Flood frequencies and sedimentation from forest watersheds. Eos, Transactions American Geophysical Union, 30(4), 567-586, 1949.

Crema, S., Llena, M., Calsamiglia, A., Estrany, J., Marchi, L., Vericat, D., Cavalli, M.: Can inpainting improve digital terrain analysis? Comparing techniques for void filling, surface reconstruction and geomorphometric analyses. Earth Surface Processes and Landforms, 45(3), 736-755, doi: 10.1002/esp.4739, 2020.

D'Agostino, V. and Marchi, L.: Debris flow magnitude in the Eastern Italian Alps: data collection and analysis, Physics and Chemistry of the Earth, Part C: Solar, Terrestrial & Planetary Science, 26, 657–663, 2001.

Marchi, L. and D'Agostino, V.: Estimation of debris-flow magnitude in the Eastern Italian Alps, Earth Surface Processes and Landforms, 29, 207–220, https://doi.org/10.1002/esp.1027, 2004.

Marchi, L., Brunetti, M.T., Cavalli, M., Crema, S.: Debris-flow volumes in northeastern Italy: relationship with drainage area and size probability. Earth Surface Processes and Landforms, 44(4), 933-943, doi: 10.1002/esp.4546, 2019.

Rickenmann, D.: Empirical relationships for debris flows. Natural Hazards 19, 47-77, 1999.

Wilford, D. J., Sakals, M. E., Innes, J. L., Sidle, R. C., and Bergerud, W. A.: Recognition of debris flow, debris flood and flood hazard through watershed morphometrics, Landslides, 1, 61–66, https://doi.org/10.1007/s10346-003-0002-0, 2004.

*We thank again very much Dr Marchi for his time spent in helping us to improve our work!*

---

## Author Comment (AC2)

Responses to the comments of Referee 2 on egusphere-2022-1494

**Editorial template**
Each comments of the reviewer are presented in the normal font.

*The responses of the authors are indented and written in italic.*

**Comments and responses**

**General comment**
With their study, the authors deal with the sediment discharge from alpine catchments in the French Western Alps. The study is based on a large number of historical measurement data, some of which go far back into the past. The authors subject the measured data to a careful plausibility check at the end of which the impressive number of 69 catchments remain for further in-depth evaluations. From a combination of these data with climatic data and digital relief analysis, various statistical techniques are used to examine those variables suspected of controlling sediment discharge from alpine catchments.

Overall, the study is well structured, based on a careful review of the literature, and written in good English. Furthermore, it must be highlighted here that very valuable data are included in the analyses, which are thus also made visible to the scientific community. Such long-term time series, especially with regard to sediment discharge, are rare and, if available, can only be put into value with great effort. Ultimately, however, these long-term time series are absolutely necessary in order to analyze statements about changes in sediment discharge (changes as a result of anthropogenic changes or caused by climate change). With the topic addressed, it has high relevance and fits very well with the focus of NHESS and I therefore strongly recommend its inclusion in the journal!

My congratulations to the authors on this work!

*We thank very much Referee 2 for this positive comment.*

**Detailed comments to the chapters:**

**Missing of the chapter Study site**
From my point of view, a chapter on the study areas is absolutely missing in the publication. At one point or another, the text refers to the variability of climate and geology, and the different slope conditions are also mentioned. However, it is difficult for the reader to understand exactly what this variability looks like! It would be desirable, for example, that a map of precipitation distribution be presented (e.g., with mean annual precipitation). Other information is of course difficult to present on maps due to the wide extent of the watersheds. Information on elevation distribution, EZG size and the different slope ratios can be found in the table in the supplement material, but here it would be worth considering whether to try to present individual important influencing variables (slope, channel lengths, vegetation cover) graphically (e.g. boxplots) rather than just using mean and median values. This would mean a good basis for the later discussion. Especially for areas with different precipitation, uncertainties in the (statistical) analyses would be more concretely discussable and explainable.

Figure 1 gives a good overview of the location of the sites (should be integrated into the chapter study sites), but due to the small size some catchments are hardly recognizable in the graph. Here I would prefer if the map would be larger in the manuscript (possibly combined with the precipitation distribution curve) and instead the photos of the catchments were moved to a separate figure.

*Good comment. We will add a short section "Study area" associated to an updated Figure 1. Here the figure and below the new section:*

[Figure]

*Figure 1. Spatial distribution of the studied sites: a) background image of elevation according to the IGN BD ALTI dabase and b) background image of mean annual rainfall according to the COMEPHORE data base (a link to access maps of each catchment is provided at the end of the paper*

*"**2.1 Study area***

*The study area is located in the northern french Alps. The studied catchments are located on a wide range of*

*mountain setting, from hills culminating below 800 m.a.s.l. at the north-west of Grenoble to torrents draining the glaciers of the Chamonix valley with summits above 4,000 m.a.s.l. (Figure 1a). This geology of the studied catchments cover both sedimentary, metamorphic and igneous rocks. The climate in the area is considered temperate without dry summer in the valleys, usually cold without dry summer above 1000~m.a.s.l. and even polar above 2000~m.a.s.l. (Beck et al. 2018). The annual mean precipitation ranges within 600 and 1800 mm with a clear influence of the relief, as well as a decreasing trend toward the east (Figure 1b) associated to the penetration into the massif of the humidity coming from the Atlantic sea."*

*However, we cannot provide a map where each catchment is visible in detail: an atlas showing the detailed maps of each catchment is accessible through a DOI in the data statement availability.*

*In addition, as suggested by Referee #2 we added a complementary figure that complete Table 2 with scatterplots and boxplots of the main input parameters versus the ratio of sediment contributing area $R_{ZP}$ which is our main explanatory variable. The updated the text in the result section to refer to this new figure.*

[Figure]

*Figure 7. Scatter plot of the main calculated variables against the ratio of sediment contributing area $R_{ZP}$ : a) catchment area A, b) channel length $L_{CE}$ , c) channel slope $S_{CE}$ , d) fan slope $S_C$ , e) Melton index M , f) daily precipitation with return period of 10 years P 24h$_{10}$, g) quantile 95% of the Connectivity Index extracted in the sediment contributing area IC95%$_{ZP}$ , h) Geological Index of D'Agostino and Marchi (2001) extracted in the sediment contributing area IG%$_{ZP}$ ; and histogram of the output variables: i) mean annual specific sediment production $V_m$/A, j) specific event magnitude with a 10 year return period $V_{10}$/A and k) reference specific event magnitude $V_{ref}$/A.*

**Material and methods**

**Precipitation**

With regard to the Precipitations section (I would use Precipitation as the heading here), the data basis remains somewhat unclear. The resolution of the reanalysis data is with 1km very good for a spatial analysis. But the question remains (since it is reanalysis data, which is a model result, at least if I am correct), if it makes sense especially with respect to the analysis of extreme events to use only pixels in a catchment area or if one should not analyze something more large-scale. The background is that the atmospheric conditions are certainly well represented by the reanalysis, but the spatial distribution is certainly not accurately predicted. In order to be able to estimate the occurrence of especially convective events for a space here, I would find it better to buffer the catchment areas a bit and thus extend the analyses a bit beyond the areas.

> *Indeed, we agree with Referee #2 that this point was not sufficiently clear. Actually, the COMEPHORE database is a combination of rain gauge and radar data and does not use atmospheric conditions to provide precipitation values. It is considered to represent adequately the spatial extent and intensity of local precipitation events (see https://doi.org/10.1007/s00382-021-05708-w and https://doi.org/10.1007/s00382-020-05558-y). This type of reanalysis is very different from a global reanalysis that assimilates mostly satellite data and results mainly from numerical weather models (e.g. ERA5-Land). A comparison of ERA5-Land and COMEPHORE in the East of the Pyrenees clearly illustrates this aspect in the following Copernicus report: https://www.spaceclimateobservatory.org/sites/default/files/2021-10/FLAude_D3.1-2.Recommendations%20on%20C3S%20data.pdf .*
> *In the study area of our study, COMEPHORE was shown to be adequate for the reproduction of the hydrological processes of small catchments (from 10 km$^2$ to 200 km$^2$), see the following technical report in French: https://hal.inrae.fr/hal-03671653  We agree that the characteristics of COMEPHORE are important to consider in our study and they will be detailed in Section 2.4.1 "Precipitation" where the following sentence will be added:*
>
>> *"The COMEPHORE product exploits ground measurements from rain gauges and radars. It is considered to represent adequately the spatial extent and intensity of intense and local precipitation events (see Appendix A in Caillaud et al., 2021, for an extensive description of its strengths and limitations)."*

**Geological index**

The weighting used is certainly suitable. The question remains, however, to what extent the geological maps used actually show bedrock and loose material. For the discharge of an area, it is ultimately not so relevant whether granite or limestone predominates as geology, but rather whether sufficient loose material is available. This can be moraine material or thicker slope debris covers. A distinction should be made, however, between bedrock and loose material. The authors

should make this a little clearer in this section, which information was really used from the geological map.

> *Very good remark: the geological map we used actually maps superficial formations, especially the many kind of quaternary deposits that are key sediment source to many torrents. As such, moraine accumulations or gullies entrenching fluvial deposits are captured. This will be specified in the text (as well as a detail on the catchments without mapped sediment contributing area). The new elements are underlined:*
>
> > *"The definition of the lithological classes was performed mainly on the basis of national geological maps which account for superficial formations as fluvial and glacial loose deposits (BD Charm-50 © BRGM, see https://www.geocatalogue.fr/Detail.do?id=4156. In catchment without mapped sediment contributing area, where even the river channel was too narrow to clearly appear between the mapped vegetation patches (an evidence of weak sediment transport activity), a minimum value of 0.5 was arbitrarily assigned.*

**Results**

From my point of view, the analysis of extreme events (magnitude and frequency) is too short. I would suggest that the authors try to include some analysis of this very important aspect in the results. Even though the data certainly have limitations in this regard and the focus of the study certainly has a different emphasis, this information would be very helpful for understanding sediment discharge. This again especially against the background of being able to discuss the uncertainties in the model result.

> *We are sorry but we are not sure to fully understand the request from Referee #2. The whole point of the statistical fit is to estimate the frequency – magnitude relationship of each basin. Each fit is shown in the Supplement in a very long Figure S5. The type of statistical fit (exponential or GPD) we used are quite standard. The paper being yet long and approaching many other topics that are newer, we prefer to stay concise on this part. Maybe we are missing the essence of the question that the Referee #2 would like to raise?*

**Discussion**

The discussion takes up important aspects of the results section, but in my view parts of the discussion are more like conclusions. In my opinion, the authors should carefully revise the text and separate the discussion from the conclusion.

**Conclusion**

Parts of the discussion can be incorporated here as Conclusion, which would also add some value to this section. So far, this part is more of a summary in my view. Here, too, I recommend that the authors carefully revise the section.

> *We respond here to the two comments above that are connected. Considering the several addition we made to the discussion, it now probably more looks like a classical discussion where the authors interpret and comment the results both in a broader perspective (comparing with other works) and with eventual more advices and comments that are not strictly "results" but more personal ideas on how to use the "results". In the new extended and rework form of the Discussion, we prefer to keep the elements in it where we explain the doubts we have on our analysis, the perspective we see to push further the work and how we suggest to reuse the results.*

> *We agree that the conclusion (that will be rather called "Concluding remarks") rather look like a synthesis but we fell it is consistent and follows well the previous Discussion. We trust the Editor to advice on whether another approach of what to put in the Concluding Remarks is necessary or not.*

Some minor suggestions (but there are maybe more):

> *Thanks a lot for these suggestions and pointing these typo!. We addressed the remarks, corrected and typos and provide responses below only if we thought it necessary.*

L23: approaches

L31: replace for instance with for example

L59: please rephrase the sentence: "The paper presents….."

L71: erratic? Perhaps better episodic?

L74: of an alluvial fan

L78: how did you assume 25%?? Is this based on expert information?

> *Yes, this is now specified*

L81: remove mean

L104-105: The sentence should be rephrased

L118: I would suggest to remove "if crude in its results"

L133: remove on

L151: what do you mean with "geometries"? Do you mean areas? Please use area also in the following sentences

L153: what is meant by "but goes essentially in the same spirit"? Please specify.

> *We will be more precise: "the definition is thus not exactly the same than that used by Haas et al., 2011; Altmann et al., 2021, who used automated threshold conditions on the land cover, the hillslope*

> *gradient, the distance to the channel and the channel slope and but goes essentially in the same spirit: identifying in mountain catchments connected, active sediment sources on aerial pictures – to identify the bare soil – and topographical maps – to check the connectivity."*

L251:remove "that"

L252: replace "corresponds" by tends to or consists of

L300: in your catchments I would think, that you mainly have bare sediments and not soil. I would suggest to use sediment or material instead of soil (also in the following text)

> *Well, as pointed by the other referee, we also sometime have bare rock. We will specify that in the paper "soil" encapsulate all of them:"i.e. unvegetated soil, sediment or rock".*

L328: I think also for this statement a map or other figure about the climate variability in a "study site" section could be helpful/necessary

L356: What do you mean with weakly active hydrosystems? Please make clear

L366: provide multiple estimations

L367: Using one single equation

L370: Debris-flow

> *We thank very much Referee #2 for his/her time helping us to improve this work!*